# Variational Autoencoding Discrete Diffusion with Enhanced Dimensional Correlations Modeling

**Tianyu Xie**[1,*]**, Shuchen Xue**[3]**, Zijin Feng**[4]**, Tianyang Hu**[5]**, Jiacheng Sun**[4]**,
Zhenguo Li**[4]**, Cheng Zhang**[1,2,†]

[1] School of Mathematical Sciences, Peking University
[2] Center for Statistical Science, Peking University
[3] Academy of Mathematics and Systems Science, Chinese Academy of Sciences
[4] Huawei Foundation Model Dept
[5] The Chinese University of Hong Kong, Shenzhen

## Abstract

Discrete diffusion models have recently shown great promise for modeling complex discrete data, with masked diffusion models (MDMs) offering a compelling trade-off between quality and generation speed. MDMs denoise by progressively unmasking multiple dimensions from an all-masked input, but their performance can degrade when using few denoising steps due to limited modeling of inter-dimensional dependencies. In this paper, we propose Variational Autoencoding Discrete Diffusion (VADD), a novel framework that enhances discrete diffusion with latent variable modeling to implicitly capture correlations among dimensions. By introducing an auxiliary recognition model, VADD enables stable training via variational lower bounds maximization and amortized inference over the training set. Our approach retains the efficiency of traditional MDMs while significantly improving sample quality, especially when the number of denoising steps is small. Empirical results on 2D toy data, pixel-level image generation, and text generation demonstrate that VADD consistently outperforms MDM baselines in sample quality with few denoising steps.

## 1 Introduction

Diffusion models (Ho et al., 2020; Song et al., 2020) have shown their remarkable successes in generative modeling of continuous objects, e.g., image (Rombach et al., 2021; Ramesh et al., 2022), audio (Kong et al., 2021; Liu et al., 2023), and video (Ho et al., 2022; Blattmann et al., 2023). Recently, diffusion models have also been extended to the discrete state space (Austin et al., 2021; Campbell et al., 2022; Sun et al., 2023; Lou et al., 2024), achieving competitive or even superior performance to autoregressive models in tasks such as Sudoku solving and code generation. A notable example is the masked diffusion model (MDM) (Sahoo et al., 2024; Shi et al., 2024; Ou et al., 2025), which works by progressively masking the dimensions of the data point towards an all-masked state in the forward process, and gradually unmasking (i.e., predicting the distributions of) multiple dimensions simultaneously in the backward process. This parallel prediction capability of MDMs offers great potentials of sampling acceleration upon autoregressive models, which typically predict one dimension at a time (Xu et al., 2024). However, the practical value of MDMs lies not just in their parallel generation capability, but critically in their ability to produce high-quality samples with minimal denoising steps—a key requirement for real-world deployment where inference speed is paramount.

Despite the inference efficiency of MDMs, their denoising distribution at each backward step is typically modeled as a product of independent categorical distributions across dimensions which may fail to capture complex inter-dimensional dependencies commonly present in real-world data. This

---

[*]Email: `tianyuxie@pku.edu.cn`
[†]Corresponding author. Email: `chengzhang@math.pku.edu.cn`

issue becomes particularly pronounced when using a small number of denoising steps, where many dimensions must be unmasked simultaneously, amplifying the impact of independence assumptions. While recent efforts (Xu et al., 2024; Liu et al., 2024) have attempted to mitigate this limitation, they require inner-loop sampling steps under guidance from pre-trained autoregressive models or correlation models, introducing additional computation costs.

In this work, we propose Variational Autoencoding Discrete Diffusion (VADD), a novel framework that enhances discrete diffusion models by incorporating a latent variable structure into the denoising distribution. This structure enables the model to implicitly capture inter-dimensional correlations, thereby improving its approximation capacity. To address the intractability of the resulting marginal distributions, we adopt the variational autoencoding framework (Kingma & Welling, 2014), jointly optimizing the denoising model and an auxiliary recognition model via a variational lower bound. We further design a specialized transformer-based architecture for VADD that retains the fast inference efficiency of traditional MDMs. Through experiments on 2D toy datasets, pixel-level image generation, and text generation, we show that VADD consistently outperforms MDM baselines—particularly in terms of sample quality with few sampling steps—highlighting the benefits of modeling dimensional dependencies.

## 2 BACKGROUND

**Masked diffusion models** Let $\boldsymbol{x}_0$ be a categorical sample with $N$ dimensions and $\boldsymbol{x}_0^i \in \{1, \ldots, V\}$ be the $i$-th dimension of $\boldsymbol{x}_0$. We further augment the state space with a special mask state $[\text{M}] = V+1$. Let $\delta_a$ denote a one-hot vector of length $V + 1$ with the value 1 at position $a$. The forward process of masked diffusion models (MDMs) is defined as $q(\boldsymbol{x}_t|\boldsymbol{x}_s) = \prod_{i=1}^N q(\boldsymbol{x}_t^i|\boldsymbol{x}_s^i)$ and $q(\boldsymbol{x}_t^i|\boldsymbol{x}_s^i) = \text{Cat}(\boldsymbol{x}_t^i; \frac{\alpha_t}{\alpha_s}\delta_{\boldsymbol{x}_s^i} + \frac{\alpha_s - \alpha_t}{\alpha_s}\delta_{[\text{M}]})$ for $s < t$, where the mask schedule $\alpha_t$ is a strictly decreasing function in $t \in [0, 1]$ with $\alpha_0 \approx 1$ and $\alpha_1 \approx 0$. Particularly, we have $q(\boldsymbol{x}_t^i|\boldsymbol{x}_0^i) = \text{Cat}(\boldsymbol{x}_t^i; \alpha_t\delta_{\boldsymbol{x}_s^i} + (1 - \alpha_t)\delta_{[\text{M}]})$ that will diffuse to an all-masked state $[\text{M}]^N$ at $t = 1$.

Given $\boldsymbol{x}_0$, the posterior distribution of $\boldsymbol{x}_s$ takes the form $q(\boldsymbol{x}_s|\boldsymbol{x}_t, \boldsymbol{x}_0) = \prod_{i=1}^N q(\boldsymbol{x}_s^i|\boldsymbol{x}_t^i, \boldsymbol{x}_0^i)$ where

$$q(\boldsymbol{x}_s^i|\boldsymbol{x}_t^i, \boldsymbol{x}_0^i) = \begin{cases} \text{Cat}(\boldsymbol{x}_s^i; \delta_{\boldsymbol{x}_t^i}), & \boldsymbol{x}_t^i \neq [\text{M}], \\ \text{Cat}\left(\boldsymbol{x}_s^i; \frac{1-\alpha_s}{1-\alpha_t}\delta_{[\text{M}]} + \frac{\alpha_s - \alpha_t}{1-\alpha_t}\delta_{\boldsymbol{x}_0^i}\right), & \boldsymbol{x}_t^i = [\text{M}]. \end{cases} \tag{1}$$

The symbol $\text{Cat}$ refers to the categorical distribution. Equation (1) inspires parametrizing the backward transitions as

$$p_{\boldsymbol{\theta}}(\boldsymbol{x}_s|\boldsymbol{x}_t) = q(\boldsymbol{x}_s|\boldsymbol{x}_t, \boldsymbol{x}_0 = \boldsymbol{\mu}_{\boldsymbol{\theta}}(\boldsymbol{x}_t, t)) = \prod_{i=1}^N q(\boldsymbol{x}_s^i|\boldsymbol{x}_t^i, \boldsymbol{x}_0^i = \boldsymbol{\mu}_{\boldsymbol{\theta}}^i(\boldsymbol{x}_t, t)), \tag{2}$$

where the denoising distribution $\boldsymbol{\mu}_{\boldsymbol{\theta}}(\boldsymbol{x}_t, t) \in \mathbb{R}^{N \times (V+1)}$ with the constraint $\sum_{j=1}^V \boldsymbol{\mu}_{\boldsymbol{\theta}}^{i,j}(\boldsymbol{x}_t, t) = 1$ and $\boldsymbol{\mu}_{\boldsymbol{\theta}}^{i, [\text{M}]}(\boldsymbol{x}_t, t) = 0$ is expected to match the posterior of the clean data $q(\boldsymbol{x}_0|\boldsymbol{x}_t)$. The $\boldsymbol{\mu}_{\boldsymbol{\theta}}(\boldsymbol{x}_t, t)$ is explicit and often trained by maximizing the continuous-time evidence lower bound (ELBO) (Shi et al., 2024; Sahoo et al., 2024)

$$\mathcal{L}(\boldsymbol{x}_0; \boldsymbol{\theta}) = \int_0^1 \mathbb{E}_{q(\boldsymbol{x}_t|\boldsymbol{x}_0)} \frac{-\alpha_t'}{1 - \alpha_t} \log p_{\boldsymbol{\theta}}(\boldsymbol{x}_0|\boldsymbol{x}_t)\mathrm{d}t \leq \log p_{\boldsymbol{\theta}}(\boldsymbol{x}_0) \tag{3}$$

for all $\boldsymbol{x}_0$ in the training set, equivalent to the training loss of any-order autoregressive models, as discussed in Ou et al. (2025).

Note that the backward transition distribution $p_{\boldsymbol{\theta}}(\boldsymbol{x}_s|\boldsymbol{x}_t)$ in equation (2) is factorizable over $N$ dimensions which helps reducing the modeling complexity of state space. However, this conditional independence structure fails to capture the inter-dimensional correlations and would inevitably introduce cumulative approximation errors. As the number of dimensions unmasked simultaneously during a backward transition is proportional to $(\alpha_s - \alpha_t)$, this drawback can be more severe under a large step size.

**Variational autoencoders** Consider the generative modeling task on a dataset $\{\boldsymbol{y}_1, \ldots, \boldsymbol{y}_M\}$, where $\boldsymbol{y}_i$ is an $N$-dimensional continuous or discrete variable. The variational autoencoder (VAE)

(Kingma & Welling, 2014) assumes a generative model $p_{\boldsymbol{\theta}}(\boldsymbol{y}, \boldsymbol{z}) = p_{\boldsymbol{\theta}}(\boldsymbol{y}|\boldsymbol{z})p(\boldsymbol{z})$, where $\boldsymbol{z} \in \mathbb{R}^d$ is a latent variable with a prior distribution $p(\boldsymbol{z})$, and a recognition model $r_{\boldsymbol{\phi}}(\boldsymbol{z}|\boldsymbol{y})$ as an approximation for the intractable posterior $p_{\boldsymbol{\theta}}(\boldsymbol{z}|\boldsymbol{y})$. The generative model and recognition model can be jointly learned by maximizing the following evidence lower bound (ELBO)

$$L(\boldsymbol{y}; \boldsymbol{\theta}, \boldsymbol{\phi}) = \mathbb{E}_{r_{\boldsymbol{\phi}}(\boldsymbol{z}|\boldsymbol{y})} \log \left( \frac{p_{\boldsymbol{\theta}}(\boldsymbol{y}, \boldsymbol{z})}{r_{\boldsymbol{\phi}}(\boldsymbol{z}|\boldsymbol{y})} \right) = \log p_{\boldsymbol{\theta}}(\boldsymbol{y}) - D_{\mathrm{KL}}\left( r_{\boldsymbol{\phi}}(\boldsymbol{z}|\boldsymbol{y}) \| p_{\boldsymbol{\theta}}(\boldsymbol{z}|\boldsymbol{y}) \right) \leq \log p_{\boldsymbol{\theta}}(\boldsymbol{y}) \quad (4)$$

for all data points $\{\boldsymbol{y}_1, \ldots, \boldsymbol{y}_M\}$. Here, $p_{\boldsymbol{\theta}}(\boldsymbol{y}) = \int_{\mathbb{R}^d} p_{\boldsymbol{\theta}}(\boldsymbol{y}, \boldsymbol{z}) \mathrm{d}\boldsymbol{z}$ is the marginal likelihood of $\boldsymbol{y}$ and $D_{\mathrm{KL}}$ is the Kullback-Leibler (KL) divergence.

A mean-field structure is often assumed for the conditional distribution $p_{\boldsymbol{\theta}}(\boldsymbol{y}|\boldsymbol{z})$, i.e., the dimensions of $\boldsymbol{y}$ are independently distributed conditioned on $\boldsymbol{z}$. As a comprehensible example, Kingma & Welling (2014) assumes the following distribution over a continuous image sample $\boldsymbol{y}$

$$p_{\boldsymbol{\theta}}(\boldsymbol{y}|\boldsymbol{z}) = \mathcal{N}\left( \boldsymbol{m}_{\boldsymbol{\theta}}(\boldsymbol{z}), \mathrm{diag}\{\boldsymbol{\sigma}_{\boldsymbol{\theta}}^2(\boldsymbol{z})\} \right), \quad (5)$$

where $\boldsymbol{m}_{\boldsymbol{\theta}}, \boldsymbol{\sigma}_{\boldsymbol{\theta}} : \mathbb{R}^d \to \mathbb{R}^N$ are two learnable neural networks. As a latent variable model, the marginal distribution $p_{\boldsymbol{\theta}}(\boldsymbol{y})$ is still capable of modeling the complex dimensional correlations by integrating out the latent variable $\boldsymbol{z}$.

## 3 METHODOLOGY

In this section, we propose Variational Autoencoding Discrete Diffusion (VADD), which extends the dimension-independent denoising distribution in MDMs by introducing a latent variable structure. This design enables VADD to capture complex dependencies across dimensions. Both the denoising distribution and an auxiliary recognition model are jointly optimized under the variational autoencoding framework. The basic idea of VADD can also be transferred to discrete diffusion models with other noise schedules (e.g., uniform distribution as the noise), which is an interesting future direction.

### 3.1 DENOISING MODEL IN VADD

Instead of parametrizing the backward transition $p_{\boldsymbol{\theta}}(\boldsymbol{x}_s|\boldsymbol{x}_t)$ as an explicit distribution, we define it as a latent variable model:

$$p_{\boldsymbol{\theta}}(\boldsymbol{x}_s|\boldsymbol{x}_t) = \int_{\mathbb{R}^d} p_{\boldsymbol{\theta}}(\boldsymbol{x}_s|\boldsymbol{x}_t, \boldsymbol{z})p(\boldsymbol{z})\mathrm{d}\boldsymbol{z}, \quad (6)$$

where $p(\boldsymbol{z})$ is the prior distribution of the latent variable $\boldsymbol{z} \in \mathbb{R}^d$ and $p_{\boldsymbol{\theta}}(\boldsymbol{x}_s|\boldsymbol{x}_t, \boldsymbol{z})$, an explicit probabilistic model, is the conditional distribution given the previous state $\boldsymbol{x}_t$ and the latent variable $\boldsymbol{z}$. Although the conditional distribution $p_{\boldsymbol{\theta}}(\boldsymbol{x}_s|\boldsymbol{x}_t, \boldsymbol{z})$ may not capture the dimensional correlations of $\boldsymbol{x}_s$, the marginal transition distribution $p_{\boldsymbol{\theta}}(\boldsymbol{x}_s|\boldsymbol{x}_t)$ is capable of doing this by integrating out $\boldsymbol{z}$. Throughout this paper, the prior distribution $p(\boldsymbol{z})$ is the standard Gaussian distribution $\mathcal{N}(\mathbf{0}_d, \boldsymbol{I}_d)$, and other multimodal distributions, e.g., Gaussian mixtures, are meaningful to explore in the future.

Inspired by the $\boldsymbol{x}_0$-prediction parameterization of MDM backward transitions in equation (2), we parametrize the conditional distribution as $p_{\boldsymbol{\theta}}(\boldsymbol{x}_s|\boldsymbol{x}_t, \boldsymbol{z}) = \prod_{i=1}^N p_{\boldsymbol{\theta}}(\boldsymbol{x}_s^i|\boldsymbol{x}_t^i, \boldsymbol{z})$ and

$$p_{\boldsymbol{\theta}}(\boldsymbol{x}_s^i|\boldsymbol{x}_t^i, \boldsymbol{z}) = \begin{cases} \mathrm{Cat}\left( \boldsymbol{x}_s^i; \boldsymbol{x}_t^i \right), & \boldsymbol{x}_t^i \neq [\mathrm{M}]; \\ \mathrm{Cat}\left( \boldsymbol{x}_s^i; \frac{1-\alpha_s}{1-\alpha_t}\delta_{[\mathrm{M}]} + \frac{\alpha_s-\alpha_t}{1-\alpha_t}\boldsymbol{\mu}_{\boldsymbol{\theta}}^i(\boldsymbol{x}_t, \boldsymbol{z}, t) \right), & \boldsymbol{x}_t^i = [\mathrm{M}]; \end{cases} \quad (7)$$

where the $\boldsymbol{\mu}_{\boldsymbol{\theta}}(\boldsymbol{x}_t, \boldsymbol{z}, t) \in \mathbb{R}^{N \times (V+1)}$, satisfying $\sum_{j=1}^V \boldsymbol{\mu}_{\boldsymbol{\theta}}^{i,j}(\boldsymbol{x}_t, \boldsymbol{z}, t) = 1$ and $\boldsymbol{\mu}_{\boldsymbol{\theta}}^{i,[\mathrm{M}]}(\boldsymbol{x}_t, \boldsymbol{z}, t) = 0$, is a deep model outputting the categorical probabilities of $p_{\boldsymbol{\theta}}(\boldsymbol{x}_0|\boldsymbol{x}_t, \boldsymbol{z})$.

Intuitively, there are multiple possible ways to recover the clean data $\boldsymbol{x}_0$ from the partially masked sample $\boldsymbol{x}_t$, reflecting the multimodality of $q(\boldsymbol{x}_0|\boldsymbol{x}_t)$. The latent variable $\boldsymbol{z}$ can be interpreted as a controller for high-level semantics, guiding the denoising model toward a specific mode of the clean data. Figure 1 provides a comprehensive example on how VADD works on 2D toy examples. We see that MDLM cannot model this correlation in one step, as suggested by the collapsed samples in the middle column. In contrast, VADD correctly captures this correlation and generates good samples from these multimodal distributions in one step.

Figure 1: One-step generation results of VADD and MDLM (Sahoo et al., 2024) on 2D examples.

Combining equation (6) and (7), the conditional distribution of $\boldsymbol{x}_0$ given $\boldsymbol{x}_t$, $q(\boldsymbol{x}_0|\boldsymbol{x}_t)$, is approximated through

$$p_{\boldsymbol{\theta}}(\boldsymbol{x}_0|\boldsymbol{x}_t) = \int_{\mathbb{R}^d} \prod_{i=1}^{N} \left[ \boldsymbol{\mu}_{\boldsymbol{\theta}}^{i,\boldsymbol{x}_0^i}(\boldsymbol{x}_t, \boldsymbol{z}, t) \mathbb{I}_{\boldsymbol{x}_t^i = [\text{M}]} + \mathbb{I}_{\boldsymbol{x}_0^i = \boldsymbol{x}_t^i \neq [\text{M}]} \right] p(\boldsymbol{z}) \mathrm{d}\boldsymbol{z}. \tag{8}$$

However, directly maximizing the ELBO $\mathcal{L}(\boldsymbol{x}_0; \boldsymbol{\theta})$ in equation (3) is no longer feasible, since $p_{\boldsymbol{\theta}}(\boldsymbol{x}_0|\boldsymbol{x}_t)$ in VADD requires marginalizing out $\boldsymbol{z}$ which is intractable (equation (8)). In what follows, we introduce an alternative tractable surrogate for the ELBO $\mathcal{L}(\boldsymbol{x}_0; \boldsymbol{\theta})$ in equation (3).

## 3.2 The variational autoencoding mechanism

Inspired by the idea of VAEs (Kingma & Welling, 2014), we consider approximating the posterior distribution of the latent variable $\boldsymbol{z}$ with an auxiliary recognition model $r_{\boldsymbol{\phi}}(\boldsymbol{z}|\boldsymbol{x}_0, \boldsymbol{x}_t) \approx p_{\boldsymbol{\theta}}(\boldsymbol{z}|\boldsymbol{x}_0, \boldsymbol{x}_t)$. Now for the $\mathcal{L}(\boldsymbol{x}_0; \boldsymbol{\theta})$ in equation (3), by treating the $p_{\boldsymbol{\theta}}(\boldsymbol{x}_0|\boldsymbol{x}_t)$ itself as a marginal likelihood conditioned on $\boldsymbol{x}_t$, the following equation gives a lower bound of $\mathcal{L}(\boldsymbol{x}_0; \boldsymbol{\theta})$:

$$\widehat{\mathcal{L}}(\boldsymbol{x}_0; \boldsymbol{\theta}, \boldsymbol{\phi}) = \int_0^1 \mathbb{E}_{q(\boldsymbol{x}_t|\boldsymbol{x}_0)} \mathbb{E}_{r_{\boldsymbol{\phi}}(\boldsymbol{z}|\boldsymbol{x}_0, \boldsymbol{x}_t)} \frac{-\alpha_t'}{1 - \alpha_t} \log \left( \frac{p_{\boldsymbol{\theta}}(\boldsymbol{x}_0|\boldsymbol{x}_t, \boldsymbol{z}) p(\boldsymbol{z})}{r_{\boldsymbol{\phi}}(\boldsymbol{z}|\boldsymbol{x}_0, \boldsymbol{x}_t)} \right) \mathrm{d}t \leq \mathcal{L}(\boldsymbol{x}_0; \boldsymbol{\theta}). \tag{9}$$

The $\widehat{\mathcal{L}}(\boldsymbol{x}_0; \boldsymbol{\theta}, \boldsymbol{\phi})$ in equation (9) is referred to as the Double Evidence Lower Bound (DELBO), as it is a lower bound of ELBO (see more details in Appendix A.1). The equality in equation (9) holds if and only if $r_{\boldsymbol{\phi}}(\boldsymbol{z}|\boldsymbol{x}_0, \boldsymbol{x}_t) \approx p_{\boldsymbol{\theta}}(\boldsymbol{z}|\boldsymbol{x}_0, \boldsymbol{x}_t)$, i.e., the recognition model perfectly fits the posterior. In our implementation, the recognition model $r_{\boldsymbol{\phi}}(\boldsymbol{z}|\boldsymbol{x}_0, \boldsymbol{x}_t)$ is a diagonal Gaussian distribution, i.e.,

$$r_{\boldsymbol{\phi}}(\boldsymbol{z}|\boldsymbol{x}_0, \boldsymbol{x}_t) = \mathcal{N}\left( \boldsymbol{m}_{\boldsymbol{\phi}}(\boldsymbol{x}_0, \boldsymbol{x}_t), \operatorname{diag}\left\{ \boldsymbol{\sigma}_{\boldsymbol{\phi}}^2(\boldsymbol{x}_0, \boldsymbol{x}_t) \right\} \right), \tag{10}$$

where $\boldsymbol{m}_{\boldsymbol{\phi}}$ and $\boldsymbol{\sigma}_{\boldsymbol{\phi}}$ are two deep models approximating the mean and standard deviation, respectively. We find that this simple implementation works fairly well in numerical studies.

Similarly to the classical VAEs, the denoising model and recognition model can be jointly optimized by maximizing the DELBO $\widehat{\mathcal{L}}(\boldsymbol{x}_0; \boldsymbol{\theta}, \boldsymbol{\phi})$ for all $\boldsymbol{x}_0$ in the training set. Since the $r_{\boldsymbol{\phi}}(\boldsymbol{z}|\boldsymbol{x}_0, \boldsymbol{x}_t)$ is a Gaussian distribution, the reparameterization trick could be utilized to compute the gradient w.r.t. $\boldsymbol{\phi}$. However, as the posterior distribution $p_{\boldsymbol{\theta}}(\boldsymbol{z}|\boldsymbol{x}_0, \boldsymbol{x}_t)$ is rather complex in the high-dimensional cases, naively maximizing $\widehat{\mathcal{L}}(\boldsymbol{x}_0; \boldsymbol{\theta}, \boldsymbol{\phi})$ can encounter with the posterior collapse issue empirically. To tackle this problem, we borrow the idea of KL annealing from Bowman et al. (2015); Yang et al. (2017) and consider the following variant of DELBO

$$\widehat{\mathcal{L}}_{\lambda}(\boldsymbol{x}_0; \boldsymbol{\theta}, \boldsymbol{\phi}) = \int_0^1 \mathbb{E}_{q(\boldsymbol{x}_t|\boldsymbol{x}_0)} \mathbb{E}_{r_{\boldsymbol{\phi}}(\boldsymbol{z}|\boldsymbol{x}_0, \boldsymbol{x}_t)} \frac{-\alpha_t'}{1 - \alpha_t} \left[ \log p_{\boldsymbol{\theta}}(\boldsymbol{x}_0|\boldsymbol{x}_t, \boldsymbol{z}) - \lambda \log \left( \frac{r_{\boldsymbol{\phi}}(\boldsymbol{z}|\boldsymbol{x}_0, \boldsymbol{x}_t)}{p(\boldsymbol{z})} \right) \right] \mathrm{d}t, \tag{11}$$

where $\lambda \in [0, 1]$ is the KL annealing weight. We summarize the training procedure of VADD in Algorithm 1.

As in other MDMs, sampling from VADD also starts from an all-masked state and recovers the clean data with the backward transition $p_{\boldsymbol{\theta}}(\boldsymbol{x}_s|\boldsymbol{x}_t)$. More specifically, given a time sequence $\{t_i\}_{i=1}^{T}$ satisfying $0 = t_0 < t_1 < \ldots < t_T = 1$, sampling from $p_{\boldsymbol{\theta}}(\boldsymbol{x}_0)$ is realized by recursively sampling

$$\boldsymbol{z}_{t_i} \sim p(\boldsymbol{z}) \text{ and } \boldsymbol{x}_{t_{i-1}} \sim p_{\boldsymbol{\theta}}(\boldsymbol{x}_{t_{i-1}}|\boldsymbol{x}_{t_i}, \boldsymbol{z}_{t_i}),$$

starting from $\boldsymbol{x}_{t_T} = [\text{M}]^N$. We summarize the sampling procedure of VADD in Algorithm 2.

---

**Algorithm 2:** Sampling from VADD

**Input:** A sequence of time steps
$\qquad 0 = t_0 < t_1 < \ldots < t_T = 1$; A
$\qquad$ transition model $p_{\boldsymbol{\theta}}(\boldsymbol{x}_s|\boldsymbol{x}_t, \boldsymbol{z})$.
**Output:** The generated sample $\boldsymbol{x}_0$.
Initialize $\boldsymbol{x}_{t_T} = [\text{M}]^N$;
**for** $i = T, \ldots, 1$ **do**
$\qquad$ Sample $\boldsymbol{z} \sim p(\boldsymbol{z})$;
$\qquad$ Sample $\boldsymbol{x}_{i-1} \sim p_{\boldsymbol{\theta}}(\boldsymbol{x}_{t_{i-1}}|\boldsymbol{x}_{t_i}, \boldsymbol{z})$;

---

---

**Algorithm 1:** Training in VADD

---

**Input:** Number of iterations $H$; Optimizer Opt; KL annealing weight $\lambda_h$; Batch size $B$.
Initialize $p_{\boldsymbol{\theta}}(\boldsymbol{x}_0|\boldsymbol{x}_t, \boldsymbol{z})$ and $r_{\boldsymbol{\phi}}(\boldsymbol{z}|\boldsymbol{x}_0, \boldsymbol{x}_t)$;
**for** $h = 1, \ldots, H$ **do**
    Sample $\{\boldsymbol{x}_0^{(b)}\}_{b=1}^B$ from the training data and $\{t^{(b)}\}_{b=1}^B$ from $\mathrm{Uniform}(0, 1)$;
    Sample $\{\boldsymbol{x}_{t^{(b)}}^{(b)}\}_{b=1}^B$ based on $\{\boldsymbol{x}_0^{(b)}\}_{b=1}^B$ and the forward masking process $q(\boldsymbol{x}_t|\boldsymbol{x}_0)$;
    Sample $\{\boldsymbol{z}^{(b)}\}_{b=1}^B$ from the recognition model $r_{\boldsymbol{\phi}}(\boldsymbol{z}|\boldsymbol{x}_0^{(b)}, \boldsymbol{x}_{t^{(b)}}^{(b)})$;
    Compute the KL annealing weight $\lambda_h$;
    Compute the Monte Carlo estimate of DELBO $\widehat{\mathcal{L}}_{\lambda_h,\mathrm{MC}}(\boldsymbol{\theta}, \boldsymbol{\phi}) := \frac{1}{B} \sum_{b=1}^B \widehat{\mathcal{L}}_{\lambda_h}(\boldsymbol{x}_0^{(b)}; \boldsymbol{\theta}, \boldsymbol{\phi})$;
    Update the parameters $\boldsymbol{\theta}, \boldsymbol{\phi} \leftarrow \mathrm{Opt}(\boldsymbol{\theta}, \boldsymbol{\phi}, -\nabla_{\boldsymbol{\theta}, \boldsymbol{\phi}}\widehat{\mathcal{L}}_{\lambda_h,\mathrm{MC}}(\boldsymbol{\theta}, \boldsymbol{\phi}))$;

---

### 3.3 DESIGN OF THE DENOISING MODEL AND RECOGNITION MODEL FOR TEXTS

An important application of VADD is text generative modeling. However, the standard transformer architecture (Vaswani et al., 2017) is not directly applicable for parametrizing the denoising model and recognition model for text generation. In this section, we address this limitation by specifically modifying the transformer architecture for both models and analyzing their inference complexities. Denote by $L$ the number of transformer blocks, by $d_e$ the dimension of token embeddings, by $d_c$ the dimension for latent variable embeddings, and by $d$ the latent dimension.

**Denoising model**   Recall that the the backward transition $q_{\boldsymbol{\theta}}(\boldsymbol{x}_s|\boldsymbol{x}_t)$ is implicitly defined by the denoising model $\boldsymbol{\mu}_{\boldsymbol{\theta}}(\boldsymbol{x}_t, \boldsymbol{z}, t)$ in equation (7), whose workflow is illustrated in Figure 2(a). To incorporate $\boldsymbol{z}$, we introduce the AdaLN layer, an affine transform depending on $\boldsymbol{z}$, that performs adaptive layer normalization (AdaLN) (Xu et al., 2019). Specifically, an inner-block MLP first takes the embedding of $\boldsymbol{z}$ as input and outputs the shift and scale that will be applied to all token embeddings in the sequence. Each of the $L$ transformer blocks consists of a self-attention layer and a feed-forward layer, both preceded by an AdaLN layer.

**Recognition model**   The recognition model $r_{\boldsymbol{\phi}}(\boldsymbol{z}|\boldsymbol{x}_0, \boldsymbol{x}_t)$ takes two sequences, $\boldsymbol{x}_0$ and $\boldsymbol{x}_t$, as input, but naively applying a transformer to them will double the computational cost. Fortunately, the observation that $\boldsymbol{x}_t$ is a partially masked version of $\boldsymbol{x}_0$ inspires a simpler model design. We introduce a binary vector $\boldsymbol{M}_t \in \{0, 1\}^N$ representing whether a position is masked ($= 1$) or not ($= 0$), and thus the pair $(\boldsymbol{x}_0, \boldsymbol{x}_t)$ can be bijectively mapped to $(\boldsymbol{x}_0, \boldsymbol{M}_t)$ which is used as model input instead. This defines the recognition model $r_{\boldsymbol{\phi}}(\boldsymbol{z}|\boldsymbol{x}_0, \boldsymbol{M}_t) = r_{\boldsymbol{\phi}}(\boldsymbol{z}|\boldsymbol{x}_0, \boldsymbol{x}_t)$, as illustrated in Figure 2(b). Similarly to the denoising model, an AdaLN layer is added before both the self-attention layer and feed-forward layer in each transformer block, with the difference that this AdaLN layer is applied exclusively to masked tokens. This operation can be implemented by first defining a learnable mask representation vector $\boldsymbol{R}_{\boldsymbol{\phi}}$ (shared across blocks), then using two inner-block MLPs on $\boldsymbol{R}_{\boldsymbol{\phi}}$ for outputting the shift and scale of the AdaLN layers in each block, and finally blocking their application to the unmasked tokens by multiplying $\boldsymbol{M}_t$.

**Remark.** *Although a more straightforward way for building $r_{\boldsymbol{\phi}}(\boldsymbol{z}|\boldsymbol{x}_0, \boldsymbol{x}_t)$ is just ignoring the dependency on $\boldsymbol{x}_t$ and only taking $\boldsymbol{x}_0$ as input, we find this will cause severe posterior collapse issue empirically, even if the KL annealing weight in the loss function (11) is carefully tuned by us.*

**Complexity analysis**   For VADD, since both $\boldsymbol{z}$ and $\boldsymbol{R}_{\boldsymbol{\phi}}$ are one-dimensional tensors, the complexity of the AdaLN layers is $O(LNd_e + Ld_cd_e)$, where $N$ is the sequence length. For the denoising model, the complexity of MLP is $O(dd_c)$, and that of the standard transformer model is $O(NVd_e + L(N^2d_e + Nd_e^2))$. Therefore, the total complexity of the denoising model is $O(NVd_e + L(N^2d_e + Nd_e^2) + LNd_e + Ld_cd_e + dd_c)$. Since the term $(Ld_cd_e + dd_c)$ is dominated, the overall complexity is thus to be $O(VNd_e + L(N^2d_e + Nd_e^2) + LNd_e)$. For the recognition model, the complexity of the AdaLN layer and transformer model remain the same as in the denoising model, while the complexity of the MLP is $O(dd_e)$. Thus, the total complexity of the recognition model is $O(VNd_e + L(N^2d_e + Nd_e^2) + LNd_e + Ld_cd_e + dd_e) = O(VNd_e + L(N^2d_e + Nd_e^2) + LNd_e)$.

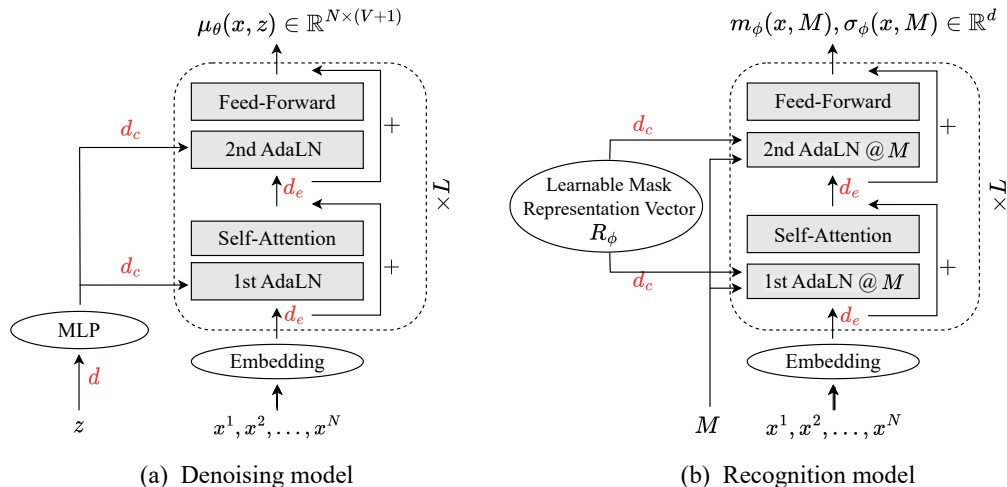

(a) Denoising model          (b) Recognition model

Figure 2: The network architecture of the denoising model and recognition model in VADD for text modeling. The feature dimensions of the tensors are marked in red font. $@M$ means that the module is only applied to the positions $i$ satisfying $M^i = 1$.

By factorizing $N$ out, the per-token complexity for both the denoising and recognition model is $O(Vd_e + LNd_e + Ld_e^2)$, where we omit the last term as it is one order smaller than the other terms.

It should be pointed out that the training cost of VADD is around $1.5\times$ that of other MDMs with only a denoising model in our implementation (see Table 4), since two models of the same size are jointly trained in VADD using the KL-annealed DELBO loss (11). As a future work for reducing the training cost, we can discard the recognition model and only optimize the generative model using alternative loss functions, e.g., score divergence. Despite the limitation of training cost, the sampling cost of VADD is the same as that of other MDMs, since the sampling procedure does not involve the recognition model.

## 4 RELATED WORKS

Several works have explored utilizing latent variable models to improve text modeling (Bowman et al., 2015; Gu et al., 2018; Kaiser et al., 2018). Recently, Kong et al. (2025) used latent variable structure for the next token prediction in autoregressive models and optimized with variational Bayes (Jordan et al., 1999), instead of the VAE amortizing over all training samples. Hayakawa et al. (2024) considered distilling the pretrained MDMs with model mixtures as the backward transition by optimizing the consistency loss. Di[M]O (Zhu et al., 2025b) distills a multi-step masked diffusion model into a one-step generator, by optimiing the the model outputs of a student model for all possible intermediate states under the help of an auxiliary model. Soft-Di[M]O (Zhu et al., 2025a) integrates soft embeddings into Di[M]O distillation, which makes one-step generators end-to-end trainable and enables straightforward application of GAN-based refinement, differentiable reward fine-tuning, and TTEO. Learnable Sampler Distillation (Fu et al., 2025) also employs a distillation approach in which a student sampler with a few steps learns to align its intermediate score trajectory with that of a high-quality teacher sampler with numerous steps. Our approach is clearly distinct from these distillation based methods as we train the latent-variable denoising model from scratch based on training data set, and do not rely on a powerful pre-trained teacher model as the initialization weight and learning target.

The posterior collapse issue is a common problem during training VAEs, i.e., the recognition model ignores the dependency on data and becomes very close to the prior distribution. From an optimization perspective, the recognition model can be less regularized by the prior through downweighting the KL divergence term, termed as the KL annealing strategy (Bowman et al., 2015; Yang et al., 2017). As the posterior collapse issue can be partially attributed to the over-complex decoder, Gulrajani et al. (2017) proposes a hybrid architecture that combines a VAE with a weaker decoder, while Kim et al. (2018) proposes to strengthen the encoder. Dieng et al. (2019) adds skip connections from the

Table 1: Empirical JS divergence (↓) between generated samples and ground truth samples, and NLL (↓) evaluated on ground truth data. JS-$T$ means sampling with $T$ steps. The empirical JS divergences are evaluated based on 100K samples.

| Model | checkerboard | | | swissroll | | | circles | | |
|---|---|---|---|---|---|---|---|---|---|
| | JS-1 | JS-5 | NLL | JS-1 | JS-5 | NLL | JS-1 | JS-5 | NLL |
| MDLM | 1.395 | 0.211 | 8.503 | 2.619 | 0.287 | 7.111 | 2.273 | 0.263 | 7.462 |
| VADD | 0.062 | **0.048** | **8.058** | 0.086 | **0.025** | **6.132** | **0.161** | **0.042** | **6.716** |

input directly to the decoder, reducing the burden on the latent variable. Kingma et al. (2016) uses normalizing flows to model a highly flexible and complex prior, allowing the posterior to capture complex data dependencies instead of collapsing to the simple prior.

## 5 EXPERIMENTS

In this section, we demonstrate the effectiveness of VADD on three tasks: two-dimensional toy examples, pixel-level image generation, and text generation. For all these tasks, we reimplement MDM using the ELBO loss function defined in equation (3), along with the same training strategy as VADD. This reimplementation serves as our main baseline, which we refer to as MDLM, a method concurrently proposed by three recent works (Sahoo et al., 2024; Shi et al., 2024; Ou et al., 2025). Unless otherwise specified, we use the linear noise schedule $\alpha_t = 1 - t$ and the linear time steps $t_i = i/T$ for discretizing the backward process. For the ELBO evaluation in VADD, we use the 1000-sample lower bound variant of DELBO (see equation (20)) as an estimate for ELBO, which is a common choice in the VAE literature (Burda et al., 2016; Vahdat & Kautz, 2020). For all the negative-likelihood-based metrics, the autoregressive results are exact likelihoods, while the diffusion results are upper bounds. The experimental details can be found in Appendix B. The official code is released at `https://github.com/tyuxie/VADD`. We emphasize that VADD is expected to achieve improved sample-quality-based metrics, such as FID score and generative perplexity, with fewer sampling steps. However, we do not expect substantial improvements in likelihood-based metrics, e.g., bits per dimension, perplexity, etc.

### 5.1 TWO-DIMENSIONAL TOY EXAMPLES

We first apply VADD to two-dimensional toy examples to provide a concise understanding of how our method works. We consider three multimodal distributions: checkerboard, swissroll, and circles. The training set consists of 100K samples generated by the `scikit-learn` package (Pedregosa et al., 2011). We rescale the data to $[0, 1)$ and discretize them with a bin width of 0.01, resulting in $V = 100$ for each dimension. Both MDLM and VADD are trained for 500 epochs with a batch size of 256 and an initial learning rate of 0.0003 (annealing according to a cosine schedule). The KL annealing weight $\lambda$ in VADD linearly increases from 0 to 1 in the first 100 epochs. The latent dimension is set to $d = 2$.

Table 1 reports the Jensen-Shannon (JS) divergence and negative marginal loglikelihood (NLL) of different methods, where we see that the VADD outperforms MDLM by a large margin on these quantitative metrics. Figure 1 and Figure 5 show the histplots of the samples generated by different methods and sampling steps, where we see that VADD is capable of generating samples that are much closer to the ground truth.

### 5.2 PIXEL-LEVEL IMAGE GENERATION

We then apply VADD to the pixel-level image generation task on the binarized MNIST (padded to $32 \times 32$, $V = 2$) and CIFAR-10 ($32 \times 32$, $V = 256$) datasets. For both datasets, the denoising model and the recognition model adopt the UNet architecture in the PyTorch implementation[1] of variational diffusion models (Kingma et al., 2021). The denoising model incorporates the latent variable $z$ by

---

[1]`https://github.com/addtt/variational-diffusion-models`

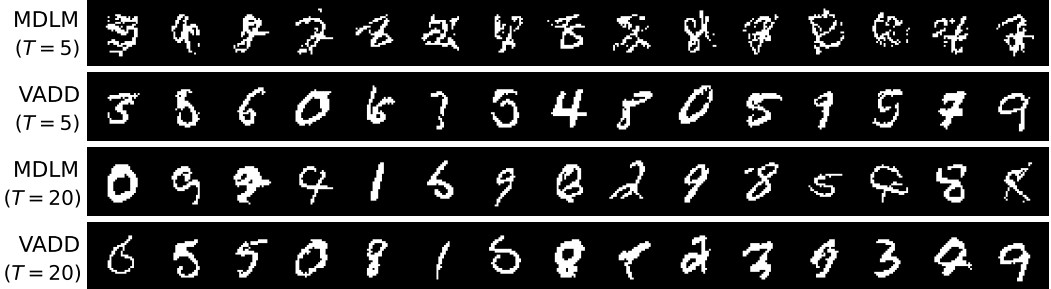

Figure 3: Non-cherry-picked samples generated by different discrete diffusion models and sampling steps on the binarized MNIST dataset.

Table 2: Test BPD (↓) on binarized MNIST and CIFAR-10. †Reproduced by us.

| Model | #Parameters | BPD |
|---|---|---|
| *Binarized MNIST (32 × 32)* | | |
| MDLM† | 2.2M | 0.075 |
| VADD (**ours**) | 2.3M | **0.063** |
| *CIFAR-10 (32 × 32)* | | |
| D3PM - $L_{\lambda=0.01}$ | 37M | 4.40 |
| MD4 | 28M | 2.75 |
| MDLM† | 32M | 2.80 |
| VADD (**ours**) | 32M | **2.74** |

Table 3: FID score (↓) with different sampling steps $T$ on the CIFAR-10 dataset. The FID score is computed with 50K images using the `clean-fid` package. †Reproduced by us.

| $T$ | MDLM† | VADD (**ours**) |
|---|---|---|
| 10 | 334.3 | **170.3** |
| 20 | 261.3 | **108.7** |
| 30 | 203.4 | **84.8** |
| 40 | 166.1 | **72.1** |
| 50 | 140.3 | **64.6** |
| 100 | 76.5 | **50.5** |

adding its embedding to the pixel embeddings in each up block and down block. The recognition model employs the siamese mechanism (Koch et al., 2015), which applies the same UNet to $x_0$ and $x_t$ and finally aggregates the outputs. The denoising model in the MDLM baseline also adopts the same UNet architecture. Both MDLM and VADD are trained for 0.2M (binarized MNIST) or 1M (CIFAR-10) iterations with a batch size of 256 and a constant learning rate of 2e-4. The KL annealing weight $\lambda$ in VADD linearly increases from 0 to 1 in the first 100K iterations.

Table 2 reports the bits per dimension (BPD) of different models. For the CIFAR-10 results in Table 2, both MDLM and VADD are trained for 2M iterations to keep consistent with that of MD4 (Shi et al., 2024). We see that the latent variable structure provides more generative modeling capacity, as evidenced by the lower BPD of VADD. It is worth noting that the BPD reduction of VADD is much more significant on the binarized MNIST dataset, as the VAE framework generally works better on low-dimensional cases (a smaller $V$).

The binarized MNIST samples generated by MDLM and VADD with different sampling steps are plotted in Figure 3. We see that VADD can generate realistic digit images with even $T = 5$ sampling steps, which is impossible for MDLM. For a direct metric for the sample quality, we report the Fréchet inception distance (FID) score between the generated images and the training set on CIFAR-10 in Table 3. We see that VADD consistently generates more realistic images than MDLM with a small number of sampling steps $T$. See more image samples in Figure 7 (MNIST) and Figure 9 (CIFAR-10) in Appendix C.2.

## 5.3 TEXT GENERATION

Finally, we test the effectiveness of VADD for unconditional text generation, aligned with the common practice of diffusion language models. We consider two widely used datasets: One Billion Word (LM1B) and OpenWebText. The denoising model and recognition model of VADD adopt the architecture in Lou et al. (2024) based on diffusion transformer, with necessary modifications for incoporating the latent variable described in Section 3.3. The reimplemented MDLM baseline also adopts the architecture in Lou et al. (2024). The sizes of these models all correspond to the

GPT-2 small scale (around 125M parameters), i.e., $T = 12$, $d_e = 784$, although our VADD may have slightly more parameters introduced by the AdaLN layers (around $6\%$). In our implementation, the denoising model and recognition model ignore the dependency on the time $t$, following Ou et al. (2025). For both MDLM and VADD, we use the AdamW optimizer with a constant learning rate of 2e-4 (after 2500 warm-up iterations) and an exponential moving average decay rate of 0.9999. The KL annealing weight in VADD increases linearly from 0 to 1 in the first 200K iterations. The results are collected after 1M iterations with a batch size of 512.

**One Billion Word**   The One Billion Word (LM1B) (Chelba et al., 2013) is a medium-sized and real-world dataset containing about 30M sentences. Following He et al. (2022); Lou et al. (2024), we use the standard train-test split, tokenize the data with the `bert-base-uncased` tokenizer, and reorganize the data into sequences with a length of $N = 128$. The latent dimension in VADD is set to $d = 128$. Two additional autoregressive baselines, Transformer-XL (Dai et al., 2019) with 0.8B parameters and OmniNet (Tay et al., 2021) with 0.1B parameters, are considered.

The perplexities on the test split of LM1B are reported in Table 5. Our VADD achieves the best test perplexity among both the autoregressive and diffusion baselines. Note that VADD with only 1M iterations even surpasses the strongest autoregressive Transformer with 5M iterations. A possible explanation is that the sequences in the reorganized dataset contain around 75% padding tokens that won't be generated, making VAE especially powerful for this relatively low-dimensional problem.

**OpenWebText**   The OpenWebText an open-source replicate of GPT-2's WebText dataset (Radford et al., 2019). Following Lou et al. (2024), we use the last 100K documents as the test split, tokenize the data with the `GPT-2` tokenizer, and reorganize the data into sequences with a length of $N = 1024$. The latent dimension in VADD is set to $d = 512$. For the zero-shot learning task, we compute the perplexities of the trained model on the test splits of six datasets: Lambada, LM1B, WikiText, AG News, PubMed, and Arxiv, following Sahoo et al. (2024).

Table 4: Training speed (it/s) of different models with a batch size of 512 on 8 H800 GPUs for the OpenWebText experiment.

| MDLM | EDLM | VADD (ours) |
|------|------|-------------|
| 2.77 | 1.74 | 1.84 |

According to the generative perplexities with 16~1024 sampling steps depicted in Figure 4, VADD consistently and significantly improves upon other MDM baselines when the number of sampling steps is small. Compared to MDLM, our VADD uses less than 50% computational cost to achieve the same sample quality. The reason lies in VADD's capability of modeling the joint distribution on multiple tokens. Table 4 reports the training speed of different models, where we see that the training speed of VADD is around $0.66\times$ that of MDLM, which is faster than $0.5\times$. Table 6 reports zero-shot perplexities on six benchmark datasets. In most datasets, our VADD model achieves similar or better zero-shot perplexities than baselines, and we emphasize that MDLM[†] uses the same architecture and training setting as VADD and is considered as the most appropriate baseline. This observation meets our expectation that VADD has more advantage in the sample-quality-based metrics than the test-likelihood-based metrics.

## 6   LIMITATIONS AND FUTURE WORKS

In the current framework, the prior distribution $p(\boldsymbol{z})$ is assumed to be a simple standard Gaussian distribution. It is the conditional denoising model $p_{\boldsymbol{\theta}}(\boldsymbol{x}_s|\boldsymbol{x}_t, \boldsymbol{z})$ that integrates the latent variable $\boldsymbol{z}$ with $\boldsymbol{x}_t$ and $t$ to model the correlations among different dimensions of $\boldsymbol{x}_s$. However, this uninformed prior can suffer from the prior hole problem, i.e., there exist regions that have high probability under the prior but low probability under the model posterior. A more informed prior structure can be $p_{\boldsymbol{\theta}}(\boldsymbol{z}|\boldsymbol{x}_t, t)$ that depends on the precedent partially masked state $\boldsymbol{x}_t$ and the time variable $t$. More complex structures, e.g., hierarchical priors (Vahdat & Kautz, 2020), can also be employed for the prior $p_{\boldsymbol{\theta}}(\boldsymbol{z}|\boldsymbol{x}_t, t)$. Advanced training techniques (Hoffman & Johnson, 2016; Aneja et al., 2020) are also applicable to alleviate the prior hole problem.

Table 5: Test perplexities (↓) on LM1B. *Reported in Sahoo et al. (2024). †Reproduced by us. The other results are reported in their original papers. All shaded model sizes correspond to GPT-2 small.

|  | Model | #Iterations | Perplexity |
|---|---|---|---|
| *AR* | Transformer-XL | - | 23.5 |
|  | OmniNet$_T$ | - | 21.5 |
|  | Transformer* | 0.5M | 22.32 |
|  | Transformer* | 5M | 20.86 |
| *Diffusion* | DiffusionBert | 1.9M | 63.78 |
|  | SEDD-Absorb | 1M | 32.79 |
|  | MDLM* | 1M | 27.04 |
|  | MDLM* | 10M | 23.00 |
|  | MDLM† | 1M | 27.70 |
|  | VADD (**ours**) | 1M | **20.53** |

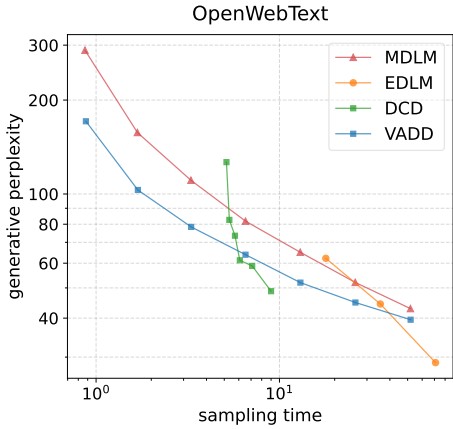

Figure 4: Generative perplexities (↓) evaluated by a pre-trained GPT-2 large model based on 256 samples on OpenWebText. All model sizes correspond to GPT-2 small.

Table 6: Zero-shot perplexities (↓) on six benchmark datasets of models trained on the OpenWebText. All models are trained for 1M iterations. Results of GPT-2 are reported in Sahoo et al. (2024). †Reproduced by us. Other results are reported in their original papers.

| Test dataset | Lambada | LM1B | WikiText | AG News | PubMed | Arxiv |
|---|---|---|---|---|---|---|
| GPT-2 (Radford et al., 2019) | 51.28 | 51.25 | 25.75 | 52.09 | 49.01 | 41.73 |
| SEDD-Absorb (Lou et al., 2024) | 50.92 | 79.29 | 40.62 | - | - | - |
| RADD-AO (Ou et al., 2025) | 49.43 | 70.71 | 35.25 | - | - | |
| MDLM† (Sahoo et al., 2024) | 49.67 | 71.03 | 35.61 | 68.57 | 42.43 | 37.92 |
| VADD (**ours**) | **47.30** | **69.71** | **34.78** | **68.00** | **40.62** | **36.39** |

## 7 CONCLUSION

In this paper, we present Variational Autoencoding Discrete Diffusion (VADD), which extends the denoising distributions in MDMs with the latent variable structure, allowing for enhanced correlation modeling over different dimensions. This advantage of VADD makes it possible for correctly generating multiple dimensions simultaneously and is especially powerful with a small number of sampling steps. The variational autoencoding mechanism is utilized to jointly optimize the denoising model together with an auxiliary recognition model. Extensive experiments on image and text generation demonstrate that our VADD consistently outperforms the MDM baselines in terms of sample fidelity under few denoising steps. Our approach is the first try of applying latent variable models and variational autoencoding mechanism to MDMs, and other expressive latent variable model designs and efficient optimization methods would be interesting future directions.

## ACKNOWLEDGEMENTS

This work was supported by National Natural Science Foundation of China (grant no. 12201014, grant no. 12292980 and grant no. 12292983). The research of Cheng Zhang was support in part by National Engineering Laboratory for Big Data Analysis and Applications, the Key Laboratory of Mathematics and Its Applications (LMAM) and the Key Laboratory of Mathematical Economics and Quantitative Finance (LMEQF) of Peking University. The authors appreciate Yu Zhu, Fangyu Ding, Ding Ding, and Haoxiong Liu for constructive discussion on this project. The authors are grateful for the computational resources provided by the High-performance Computing Platform of Peking University.

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

**LLM Usage**  In the preparation of this manuscript, LLM was used to polish grammar, style, and readability of the text.

## A  DERIVATIONS OF THE DELBO

### A.1  THE LOWER BOUND OF ELBO

Let $r_\phi(z|x_0, x_t)$ be a recognition model with support on $\mathbb{R}^d$, which can simply realized by assuming a Gaussian family for $r_\phi(z|x_0, x_t)$. We have

$$\int_{\mathbb{R}^d} \frac{p_\theta(x_0|x_t, z)p(z)}{r_\phi(z|x_0, x_t)} r_\phi(z|x_0, x_t)\mathrm{d}z = \int_{\mathbb{R}^d} p_\theta(x_0|x_t, z)p(z)\mathrm{d}z = p_\theta(x_0|x_t). \tag{12}$$

Using Jensen's inequality and noting that $\log()$ is a concave function, we have

$$\log p_\theta(x_0|x_t) = \log \int_{\mathbb{R}^d} \frac{p_\theta(x_0|x_t, z)p(z)}{r_\phi(z|x_0, x_t)} r_\phi(z|x_0, x_t)\mathrm{d}z \tag{13}$$

$$\geq \int_{\mathbb{R}^d} \log\left(\frac{p_\theta(x_0|x_t, z)p(z)}{r_\phi(z|x_0, x_t)}\right) r_\phi(z|x_0, x_t)\mathrm{d}z \tag{14}$$

$$= \mathbb{E}_{r_\phi(z|x_0, x_t)} \log\left(\frac{p_\theta(x_0|x_t, z)p(z)}{r_\phi(z|x_0, x_t)}\right). \tag{15}$$

The inequality (14) in holds if and only if $r_\phi(z|x_0, x_t) = p_\theta(z|x_0, x_t)$. By noting that $-\frac{\alpha'_t}{1-\alpha_t} > 0$, we have

$$\widehat{\mathcal{L}}(x_0; \theta, \phi) := \int_0^1 \mathbb{E}_{q(x_t|x_0)}\mathbb{E}_{r_\phi(z|x_0, x_t)} \frac{-\alpha'_t}{1-\alpha_t} \log\left(\frac{p_\theta(x_0|x_t, z)p(z)}{r_\phi(z|x_0, x_t)}\right) \mathrm{d}t \tag{16}$$

$$\leq \int_0^1 \mathbb{E}_{q(x_t|x_0)} \frac{-\alpha'_t}{1-\alpha_t} \log p_\theta(x_0|x_t)\mathrm{d}t \tag{17}$$

$$= \mathcal{L}(x_0; \theta). \tag{18}$$

The inequality (17) holds if and only if for any $t \in [0, 1]$ and $x_t \sim q(x_t|x_0)$, it holds that $r_\phi(z|x_0, x_t) = p_\theta(z|x_0, x_t)$.

Finally, we conclude that the following chain of lower bounds:

$$\widehat{\mathcal{L}}(x_0; \theta, \phi) \leq \mathcal{L}(x_0; \theta) \leq \log p_\theta(x_0). \tag{19}$$

### A.2  THE MULTI-SAMPLE VARIANT OF DELBO FOR ESTIMATING ELBO

For previous MDMs, e.g., Sahoo et al. (2024); Shi et al. (2024); Ou et al. (2025), the ELBO $\mathcal{L}(x_0; \theta)$ is reported as a surrogate of the intractable likelihood $\log p_\theta(x_0)$ and then used to compute the perplexities, bits per dimension, etc. However, for VADD, computing the ELBO is intractable as the integration on $z$ is intractable. We consider the following multi-sample variant of the DELBO, termed $K$-sample DELBO,

$$\widehat{\mathcal{L}}_K(x_0; \theta, \phi) = \int_0^1 \mathbb{E}_{q(x_t|x_0)}\mathbb{E}_{z^1,\ldots,z^K \sim r_\phi(z|x_0, x_t)} \frac{-\alpha'_t}{1-\alpha_t} \log\left(\frac{1}{K}\sum_{i=1}^K \frac{p_\theta(x_0|x_t, z^K)p(z^K)}{r_\phi(z^K|x_0, x_t)}\right) \mathrm{d}t. \tag{20}$$

The $K$-sample DELBO satisfies

$$\widehat{\mathcal{L}}_K(x_0; \theta, \phi) \leq \widehat{\mathcal{L}}_{K+1}(x_0; \theta, \phi) \leq \mathcal{L}(x_0; \theta). \tag{21}$$

Moreover, by the strong law of large numbers, it holds that

$$\lim_{K\to\infty} \widehat{\mathcal{L}}_K(x_0; \theta, \phi) = \mathcal{L}(x_0; \theta). \tag{22}$$

In all of our experiments, we use $\widehat{\mathcal{L}}_{1000}(x_0; \theta, \phi)$ as an approximation for the $\mathcal{L}(x_0; \theta)$. For the Monte Carlo sampling on $(x_t, t)$, we sample 100 independent pairs $(x_t, t) \sim q(x_t|x_0) \cdot U[0, 1]$ to estimate the $\widehat{\mathcal{L}}_{1000}(x_0; \theta, \phi)$.

# B  DETAILS OF EXPERIMENTAL SETUP

## B.1  TWO-DIMENSIONAL TOY EXAMPLES

We consider three multimodal distributions: checkerboard, swissroll, and circles. The training set consists of 100K samples generated by the `scikit-learn` package (Pedregosa et al., 2011). For checkerboard, nrows=2 and ncols=2; for the swissroll, noise=0.2; for the circles, noise=0.02 and factor=0.5. We rescale the data to $[0, 1)$ and discretize them with a bin width of 0.01, resulting in $V = 100$ for each dimension. The latent dimension is set to $d = 2$.

The denoising model $\boldsymbol{\mu_\theta}(\boldsymbol{x}_t, \boldsymbol{z}, t)$ takes a integer-valued sample $\boldsymbol{x}_t$, and a vecotr $\boldsymbol{z}$, and a scalar $t$ as inputs. All the activation functions in MLPs are ELU unless other specified. $\boldsymbol{\mu_\theta}(\boldsymbol{x}_t, \boldsymbol{z}, t)$ consists of the following two steps:

- **Embedding**. The embedding of $t$, $\mathrm{emb}(t)$, is the output of positional sinusoidal embedding (Vaswani et al., 2017) and a following MLP with channel widths [1024, 512, 512]. The embedding of $\boldsymbol{z}$, $\mathrm{emb}(\boldsymbol{z})$, is the output of an MLP with channel widths $[d, 512]$. The embedding of $\boldsymbol{x}$, $\mathrm{emb}(\boldsymbol{x})$, is the output of an embedding module with embedding dimension 512.

- **Readout**. We then sum up $\mathrm{emb}(t), \mathrm{emb}(\boldsymbol{z}), \mathrm{emb}(\boldsymbol{x})$ and apply an MLP with channel width $[512, 512, 512, 512, 512, V]$.

The recognition model $r_{\boldsymbol{\phi}}(\boldsymbol{z}|\boldsymbol{x}_0, \boldsymbol{x}_t)$ takes two integer-valued vectors $\boldsymbol{x}_0, \boldsymbol{x}_t$ and a scalar value $t$ as input and output the vector-valued mean and standard deviation of $\boldsymbol{z}$. We consider a Siamese scheme to incorporate the two inputs $\boldsymbol{x}_0, \boldsymbol{x}_t$.

- **Embedding**. The embedding of $t$, $\mathrm{emb}(t)$, is the output of positional sinusoidal embedding (Vaswani et al., 2017) and a following MLP with channel widths [1024, 512, 512]. The embeddings of $\boldsymbol{x}_0$ and $\boldsymbol{x}_t$, $\mathrm{emb}(\boldsymbol{x}_0)$ and $\mathrm{emb}(\boldsymbol{x}_t)$, are the output of the same embedding module with embedding dimension 512.

- **Readout**. We then apply MLP with channel width [512, 512, 512, 512, 512, 512] on $\mathrm{emb}(t)+\mathrm{emb}(\boldsymbol{x}_0)$ and $\mathrm{emb}(t)+\mathrm{emb}(\boldsymbol{x}_t)$, and obtain the outputs $\mathrm{out}_0$ and $\mathrm{out}_t$. Finally, we compute the average of $\mathrm{out}_0$ and $\mathrm{out}_t$, and apply an MLP with channel width $[512, 512, 2d]$.

We use the Adam optimizer with momentum parameters $(\beta_1, \beta_2) = (0.9, 0.999)$ and weight decay 0. The initial learning rate is 3e-4 and decreases according to a cosine annealing schedule. The KL annealing weight increases linearly from 0 to 1 in the first 100 epochs. The batch size is 256. The total number of training epochs is 500. The experiments are run on a single A100 40G GPU.

## B.2  PIXEL-LEVEL IMAGE GENERATION

The binarized MNIST data binarized MNIST ($32 \times 32$, $V = 2$) are obtained by binarizing the grayscale MNIST data with a threshold of 0.5 and padding 2 zeros on each side of the image. CIFAR-10 ($32 \times 32$, $V = 256$) can be directly obtained in its original form without any transformation.

The denoising model and the recognition model adopt the UNet architecture in the PyTorch implementation (https://github.com/addtt/variational-diffusion-models) of variational diffusion models (Kingma et al., 2021).

- **Binarized MNIST**. For both the recognition model and the denoising model, the numbers of up blocks and down blocks are 8. The embedding dimension is 64. The pixel embedding dimension is 32. The dropout rate is 0.1. The number of normalization groups is 16. The number of attention heads is 1. The latent dimension is 64.

- **CIFAR-10**. For both the recognition model and the denoising model, the numbers of up blocks and down blocks are 32. The embedding dimension is 128. The pixel embedding dimension is 32. The dropout rate is 0.1. The number of normalization groups is 32. The number of attention heads is 32. The latent dimension is 128.

We make the following modifications for the denoising model and the recognition model.

- **Denoising model**. A special token [M] is also added to the table of the pixel embedding module. We transform the latent variable $z$ into an embedding $\text{emb}(\boldsymbol{z})$ with MLPs and add it to the embeddings of all pixels in each up block and down block.

- **Recognition model**. Borrowing the idea of the siamese mechanism, a standard UNet takes $(\boldsymbol{x}_0, t)$ and $(\boldsymbol{x}_t, t)$ as inputs, and outputs two tensors $\text{out}_0$ and $\text{out}_t$. We compute the average $(\text{out}_0 + \text{out}_t)/2$ and downsample it to a $1 \times 1 \times 2d$ tensor with down convolution blocks. This $1 \times 1 \times 2d$ tensor will give the mean and standard deviation of $r_{\boldsymbol{\phi}}(\boldsymbol{z}|\boldsymbol{x}_0, \boldsymbol{x}_t)$.

We use the AdamW optimizer with $(\beta_1, \beta_2) = (0.9, 0.99)$, a weight decay of $0.01$, and a constant learning rate of 2e-4. The batch size is 256. The gradient is clipped to a maximum norm of 1. The KL annealing weight linearly increases from 0 to 1 in the first 100K iterations. The total number of iterations is 200K for binarized MNIST and 1M for CIFAR-10. The exponential moving average decay rate is 0.9999, and the update frequency is 1. The experiments of binarized MNIST are run on a single A100 40G GPU. The experiments of CIFAR-10 are run on 8 A100 40G GPUs.

### B.3 TEXT GENERATION

For the One Billion Word dataset, we firstly detokenize the texts following Lou et al. (2024). We then tokenize the texts with the `bert-base-uncased` tokenizer, following He et al. (2022); Lou et al. (2024). We pad and truncate the sequences to a length of 128.

For the OpenWebText dataset, we firstly detokenize the text following Lou et al. (2024). We then tokenize the texts with the `GPT-2` tokenizer. We concatenate and wrap them to a sequence length of 1024, including a `BOS` and a `EOS` token as the first and last token of the sequence. We use the last 100K documents as the validation split.

The network architecture and inference complexity of the denoising model and the recognition model have been elucidated in Section 3.3. The model size corresponds to the GPT-2 small scale, i.e., 12 transformer blocks, an embedding dimension of 768, and 12 heads in each attention layer.

The optimizer is AdamW with $(\beta_1, \beta_2) = (0.9, 0.999)$, a weight decay of $0.0$. The learning rate is a constant 3e-4, with a warmup phase of 2500 steps. The batch size is 512. The gradient is clipped to a maximum norm of 1. The KL annealing weight linearly increases from 0 to 1 in the first 100K iterations. The total number of iterations is 1M. The exponential moving average decay rate is 0.9999, and the update frequency is 1. The experiments on One Billion Word are run on 8 A100 40G GPUs, and the experiments on OpenWebText are run on 8 H800 GPUs.

## C ADDITIONAL EXPERIMENTAL RESULTS

### C.1 TWO-DIMENSIONAL TOY EXAMPLES

Figure 5 shows the histplots of the samples generated by MDLM and VADD. We see that VADD generates samples that are more close to the ground truth with both 1 and 5 sampling steps. We also visualize the samples under $1, 5, 10, 20$ sampling steps of VADD in Figure 6. We see that VADD produces very similar samples in this 2D toy experiment.

### C.2 PIXEL-LEVEL IMAGE GENERATION

Figure 7 provides the binarized samples generated by VADD. The training and test negative DELBO curves for VADD and MDLM is plotted in . We see that for both models, the training loss decreases steadily, and VADD will outperform MDLM in the end. We provide CIFAR-10 samples generated by VADD with different sampling steps in Figure 9.

We also consider a continuous DDPM (Ho et al., 2020) baseline, which is implemented by the `diffusers.DDPMPipeline` module with the `google/ddpm-cifar10-32` pre-trained checkpoint. As shown in Table 7, VADD consistently outperforms the continuous diffusion baseline DDPM across all sampling steps on CIFAR-10. At 10 steps, VADD achieves an FID of 170.3 compared to 298.7 for DDPM, demonstrating a 43% improvement. These results demonstrate that discrete diffusion can achieve competitive or superior sample quality compared to DDPM, particularly in the few-step regime. However, it should be acknowledged that the pixel-level discrete modeling of images

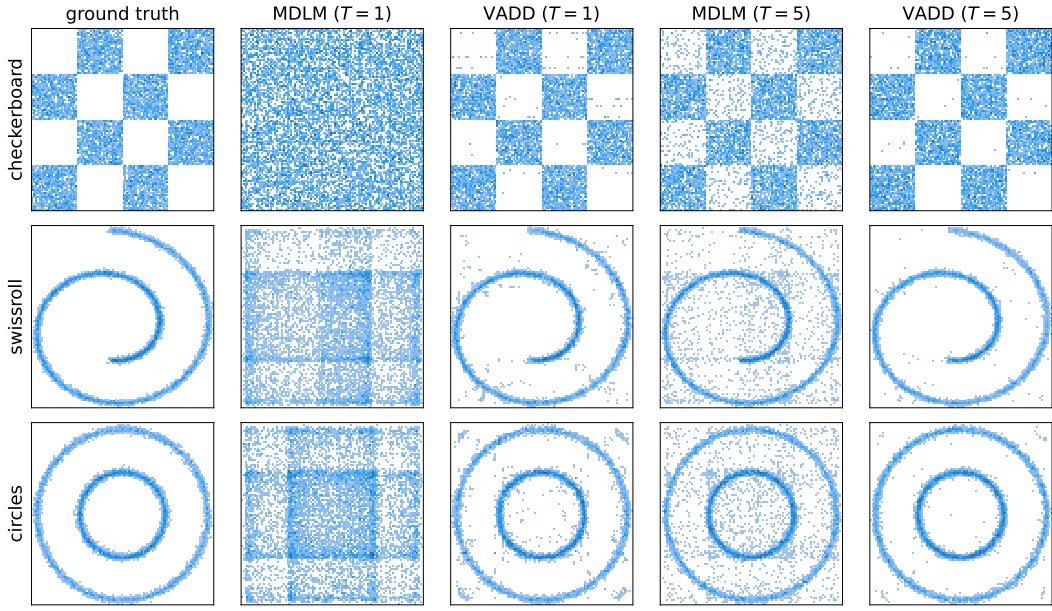

Figure 5: Histplots of the ground truth and the samples generated from different models and sampling steps on the 2D toy example.

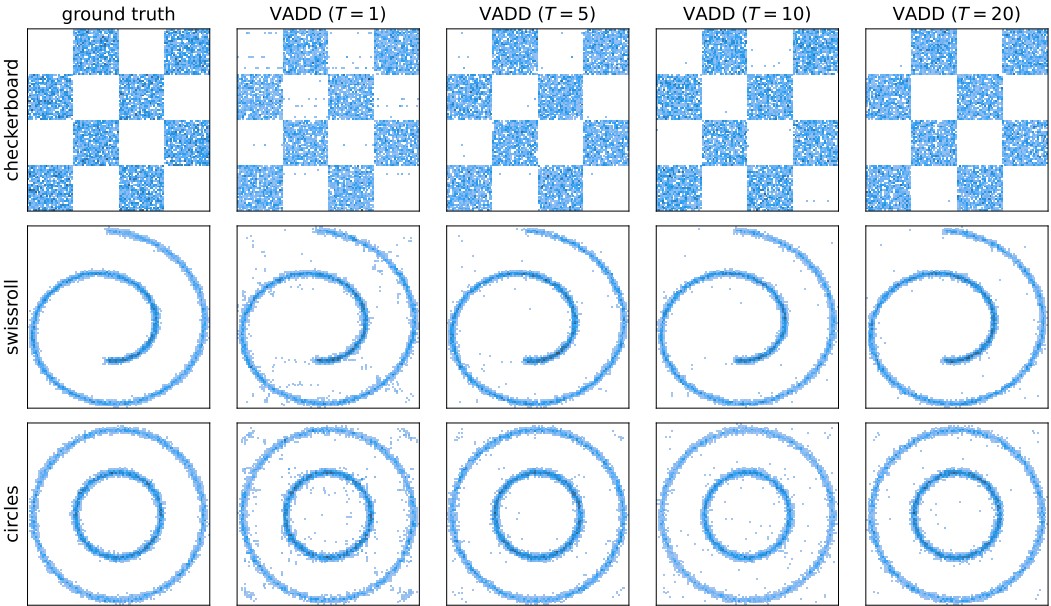

Figure 6: Histplots of the ground truth and the samples generated from VADD with various sampling steps on the 2D toy example.

is quite hard, and the FID scores of MDM-based methods could lag behind other state-of-the-art continuous modeling methods by a large margin.

## C.3 TEXT GENERATION

Table 8 shows the test perplexities on the OpenWebText dataset. To inspect how the number of particles will affect the estimate of ELBO, we also perform an ablation study on the number of

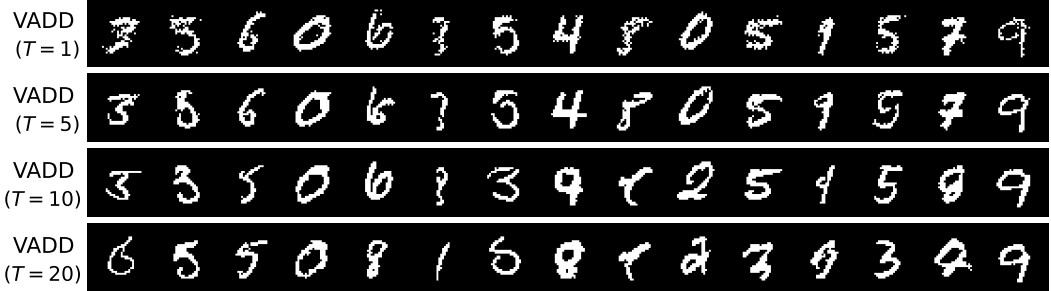

Figure 7: Samples generated by VADD with different sampling steps on the binarized MNIST.

Table 7: FID score (↓) comparison between VADD and DDPM with different sampling steps $T$ on the CIFAR-10 dataset. VADD consistently achieves lower FID scores across all sampling steps. FID is computed with 50K images using the `clean-fid` package.

| $T$ | 10 | 20 | 30 | 40 | 50 | 100 |
|---|---|---|---|---|---|---|
| VADD (**ours**) | **170.3** | **108.7** | **84.8** | **72.1** | **64.6** | **50.5** |
| DDPM | 298.7 | 239.1 | 187.2 | 153.6 | 131.0 | 77.2 |

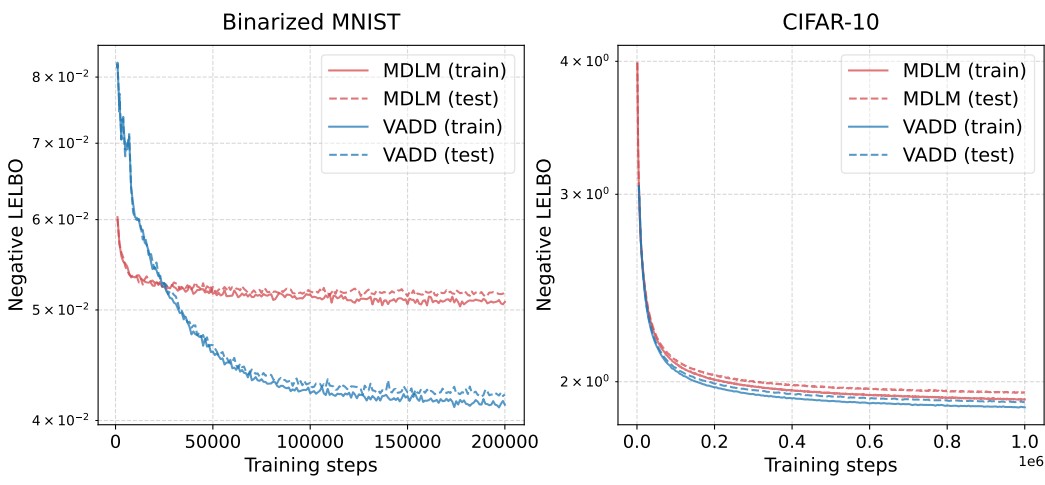

Figure 8: Training and test negative DELBOs on the pixel-level image generation task.

particles $K$ (see the $K$-sample DELBO (20)). Note that this estimate is identical to DELBO when $K = 1$. We see that the ELBO estimate gradually improves as $K$ increases.

Zheng et al. (2025) points out that the generative perplexity might be sensitive to the precision of categorical sampling. To this end, we compute the generative perplexity using fp64 in Table 9, where we see that VADD consistently outperforms MDLM.

The generated samples from VADD trained on OpenWebText with 128, 256, 512, and 1024 sampling steps are shown in Figure 10-13, respectively. The categorical sampling is under fp32 precision.

# D   BROADER IMPACT

This paper presents work whose goal is to advance the field of machine learning. There are many potential societal consequences of our work, none of which we feel must be specifically highlighted here.

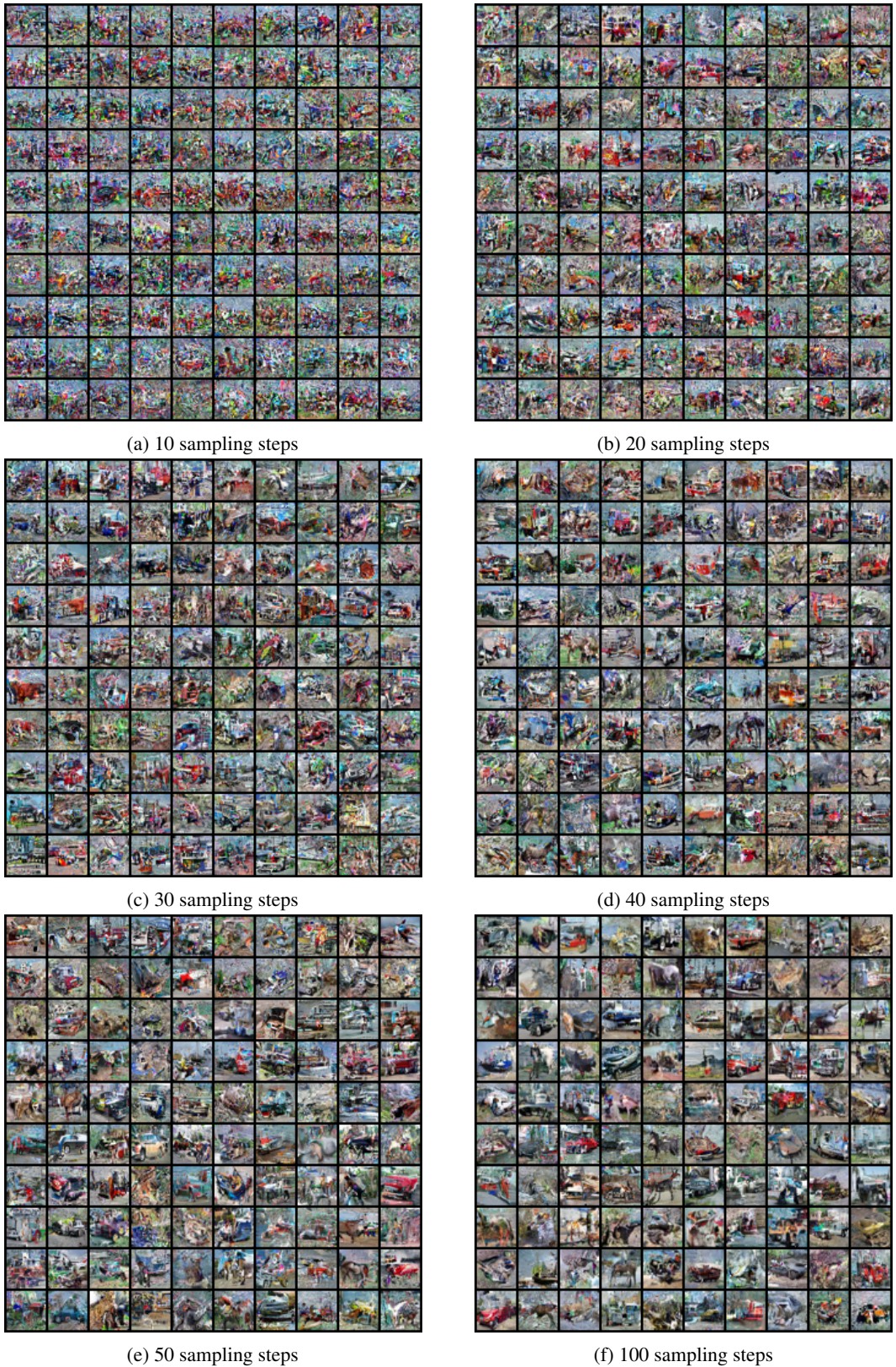

(a) 10 sampling steps

(b) 20 sampling steps

(c) 30 sampling steps

(d) 40 sampling steps

(e) 50 sampling steps

(f) 100 sampling steps

Figure 9: CIFAR-10 samples generated from VADD with various sampling steps.

<|endoftext|> the symptoms.

New thoughts are possible, and experiences that will help keep a person of heightened focus, understanding and still committed to making them better and faced with a distraction. One of these persons may have even a good model of understanding for him, but there is always something else that allows the person to express his interests.

Sometimes you're over it that way, then it goes away and over it I think, one day it can make everything go away and tap open again, even if you feel the bridge almost comfortably, not going to a situation that suddenly forces you to move, but more difficult things take place that you couldn't both before, how you wanted to have family person with you when it happens, it's for so much help, as a privilege, and me both, even clandestinely, while being given authentic adult guidance, can be frustrating and disengaged, and when life turns upside down, you are on your own as we're explaining the nuances of emotions. they take place without someone's presence.

The worst of all, though, is going to therapy the next morning, from my friend Volok Wuska. He dealt with his abusive abusive spouse for several months, assuming the gaze of his therapist but not directly expressing his own grieving feelings. He understands how you experience in healing, and effective language can be replaced with one that treated him a great deal of anxiety and anxiety. His spouse were afraid of their words, and for a day when they found themselves in grief, and being focused for a day afterward, simply for something to say. This case with a family issue and with a family business at the same time and such, the only reason they didn't get through it was therapy helped calm down that source of tension. See, the biggest benefit this therapy reveals is that the listener and the wounded fully continue to heal.

The "them economics" of friendship is gift. At best, we know we have a better understanding, and there is some greater freedom for each other because, even if we talk we're still around, we can forever enjoy all life work together with a loved one in a reciprocating meaningful but marginal if functional way. You can drink like your family relatives or colleagues who don't respect your dignity and again, "What do you screw me on the back for doing this for?"

The adult knows his feelings are over in uncomfortable circumstances, but it keeps him alive and likely to approach things that way, and allows people that learning means good and achieving good things, without doing or trying to limit our experiences or attain them. Mindful people try to remain hardy because we have our ideas, evidence and move on flow. But information, emotions and linguistics have time to be loose everywhere. If we want communicate more successfully, we have to try to avoid these situations, we have to follow you, we have to find a way of projecting, where knowledge, creativity and understanding is enjoyed from that outside mind, and what people usually throw away in the hope that life will keep hurting them even after arriving from people who might hurt them. We talk about way too much in our heads, or we'll lose a few jobs in that we actually practice over and over again, because instead of what we're feeling from the outside and getting a handle to it this urge becomes to believe something and take action. This is thinking that you can turn it off, and then trained to Act itself, when your enemy changes how he takes detains you, and I still believe in my part that getting a paycheck is worth it if you care. That said, a kackeating direction in his particular direction means that a person who of course has the lisp, and beliefs, will be acting on his opinions. If he isn't able to do that, he probably is not; we should go and wait for him to be listened to, but if you have something left out because of your rudeness and see it as reproach and suspend it, don't coach yourself again with rudeness until your board of boosters, and take every opportunity to run other people you have to tell about it.

The bottom line? You choose you will, and people in your position will respect to do that. A person you choose to leave on your side of the equation should know that you aren't supposed to take control, but you can interfere with simple tasks like toasting and photographing photos, so there may be less danger, if your statement disrupts expectations on that side.

Break down the Starbucks bar during dinner upon a your friends at Baniac a table and they give them one of their shots.

Then you go to the Arizona Tea and take it with them. Or, somewhere in the future, you are certainly going to walk away and judge any young person relating the perceptions of your lookalike and theories to reality because<|endoftext|>

Figure 10: Text sample generated by VADD trained on OpenWebText with 128 sampling steps.

<|endoftext|> to friends while I was out, so I gave out gifts of books and trying to discover new things and new goals everyday. I coaxed them around them, and done some things that I felt was important in my life; for one was to make both of my kids very happy. I didn't want both of my kids having a parent that I would be able of without making me self happy and catching me with my child happy and so happy that all that I wanted success, private happiness, was, ultimately, about internalizing my relationship. I still have a lot of my secrets, but as of today in this journey, these are what myself believe to be real good secrets about the whole experience. Being a woman was telling what was true to me; storytelling about the things that made me serve a woman and make myself happy, and how I truly wanted a little more. When I have left a positive or negative mindset I choose to not be the one I feel one day want to be, sometimes; it's not necessarily right in the every single moment. I did what I wanted, and I was more passionate about the work I love, something my kids took care of and I cared about my children, and I honestly thought about my life and my future. This is just a part of me that's what it's really like to be happy about your life.

One change for me is when I truly knew about how everything matters. I started to not talk about meeting guys and move on, just talk of how I liked what I was going to do. I felt more commitment was important; what I wanted to talk about was to mentally show that I loved being around someone, rather than making a bunch of money; I would spend money on my life and really care about it. That I also made me feel confident I was masculine and that women were not; being single made me feel accepted and a better person to be masculine, and there are many more ways to make that connection; doing some of these things with real men will make this process easier through the fact that they have the women encouraging them to both be themselves and to help them relate to the other. And of course, I think men need a strong relationship as much as possible, and women need to speak well of each other. We have to ask why we can't take care of ourselves and stay far behind. We have to be compassionate, because that's if we see how in others can go beyond our comfort and to be more positive and true to our self. And that's just as important that we believe that our friend can serve their friend; because we. can.

Being involved with ourselves, with a relationship that can work, is to love the things we want, and to like the things we like to think. Having more positive things built on us and our sense of self makes it really tough for us to control over it. Because men are our coaches when it comes to be much more than just single and married, we really need more help from a woman that is still alive and well; making her presence that important away. But being who we are, we've all been able to achieve less of our goals, maybe we desire to look even prettier but men would do these things for us; this isn't a time when men are coming from who we are, who we are and how we behave. Being single, and opening up those opportunities is about finding potential friends, and shining light into our inner lives. It is about showing off where we are in a positive way.

That's why I felt like I got all the stuff I wanted without a woman. There are more layers to a failed relationship, where there may have been, and even during first transition time, it's difficult to imagine just how far a woman has come within society. Women also tend to have more negative emotional brought-ups. Some depression is depression, depression due to their sexuality is leading state of depression, and a symptom of that alone lead to them. It's also to those who have generally high expectations and positive life expectations. Don't imagine how a woman would has said to you to change this or break you. When health comes where gender is accepted and correct. Being in person is person you truly want its love as them.

She's the guy the woman wrote about in order to protect people her mom knows are used just to press the power of self-important ideas, to shut off their entire personality, to deny themselves resentment at once for ever.

More people are accusing getting these attitude that's out in person; the women in the original author post reported believing that someone encouraged her into being a bisexual than her. Before the ex-girlfriend went into hiding in the other post, now the other post is saying that one of these threats is her being openly gay started being a cook box Oreo. <|endoftext|>

Figure 11: Text sample generated by VADD trained on OpenWebText with 256 sampling steps.

<|endoftext|> for an average 11 hour period. I used to call attention to him and people that he looked up to in his area, so I would seek him out. Despite the fact I received so much attention afterward, I experienced some of my past urges. On the first day at Lee and Barron's, I came home with a little worry. I'm not perfect. I wanted it to be good to be with him until 4PM every day. Everything he has done, I feel that I deserve. If I could tolerate my mistakes and understood my strengths and weaknesses, as well as help him get my lunch out of reach, even then that I wondered why I could not apologize, I should go further, thinking that if I did so, I would see again that I had changed nothing. Instead, I thought I could get there the next morning during a day to find out what was needful, and given that I was both tired and my rest, I'd rather appreciate that he wants our help, who wants him to help us. He came back and saved me in those moments knowing that I understood that I'm not being taken to an advisor anymore, and hoped that I could refine my advisor approach too. Once again, we were on the same page, he said the message that at its heart, no one should ever fail our business together. He brought the two together, he effectively told me he was working out and happy to listen upon my decisions in response to.

It wasn't long, however, that I caught myself caught up in his social decision-making, and also in his personal communication. Now, he speaks the truth, and speaks for the complexities that he is faced with today. Let's say it in words – his version of the mantra "the word in a hand can change the mind." Then, especially shortly after he walked away in office, I seem to have gotten off empathy on my own, I didn't interact like this in life and I couldn't change it, that was the point of the Presidential campaign. While I proud when policy and people replace the very different decision of forthcoming, I knew that I hadn't. Anyone that has been in office, means you have to do the things we want to do, in order to continue on in the world, I know that I must give these instructions, act face to face with them, and never act outside the confines of office.

I had been intimidated by the base I built, and I was definitely not confident. At the time, this is who I hoped to become. But, I had to let it be because everything was going so well. As it did, I not only knew how easy it was, but I did enjoy it on some level. And yet, I was afraid that for someone that I was supposed to be to be expected, I had to cast an eye to his life and learn on what he was now doing. When someone is doing well with a certain outlook, they will only make improvement sometimes. People never deserve to be dependent on and have to learn from on some level.

Despite this transition to office, I tried staying away. Maybe I wanted a speaking farewell. Instead, I left politics and put politics behind politics. I had a parent that seemed quite a brave choice to choose. It therefore came as a surprise that while at my turn of events, I came home showed a lesson the most of people that I know should be fired as myself needed to protect myself, it needs to be alright. My time at school in Berkeley is not easy to reflect upon, it's education itself, but I did learn something in it.

I learned how my friends here in this land trust me, they sent me out on this campus as a stranger with complete sincerity. Everyone opened their eyes, I learned the simple contact's skills that garner respect of lots of others. From an identity I gained the respect and respect to become part of such a large network. As I waited in line during the 3-hour wait alone, I wondered if everyone knew what happened and just wanted to find out how to accept me that no matter how positive the outcome. Things like this to be sure was a painful situation to live in.

I cannot attest to how difficult it is to learn, it's hard to imagine yourself when having interactions with people in life. I am fully determined that the higher marks my period of experience are through this college age always, that I am able to operate in the very different environment beneath the narrator's walls of a college where playing game with the people we have studied is very different. That said, for me the result showed that I and the students in the group do a pretty good job of keeping disdisappearing students up, calling them out and making some decisions based on them in theory. So, I can still get to learn what to say to them, remembering<|endoftext|>

Figure 12: Text sample generated by VADD trained on OpenWebText with 512 sampling steps.

<|endoftext|> situations can you have with yourself in terms of expectations? "Well, now we have to decide what we are going to do next. "What months have passed"? Sometimes our usual selfless conversations here are more deliberate. Be aware that on either side of your sleep schedule are constantly working at your social media, and getting a lot of value info about your job status tomorrow, where you'll be working, and where you might be with your friends. Don't think directly to yourself this way: "Given the amount of people you interact with, the so many experiences, how are you looking around to plan for your future and for your dreams, that may be a much easier to see when you go, and therefore making good actions, and the repercussions, and all of the advice that we've always accepted, is not inevitable.

· From what I know, that things are going to be done in line for longer than most people realize. Maybe that's one thing. We don't need to plan those small steps. Sometimes we need to slow things down because then anticipate the steps we've already taken. If, one reason this takes a lot of practice is to bear with yourself. Because we have a lot of time to make things happen. We don't have to wait for things out. No, we have to stand by about the things we've already done. Let's reflect on what went before here past today, into the future. When your List comes up, you have to put those worries ahead of you and go and listen while sorta.

· Even when you go looking for those things that'll be in there, at least behind the scenes, things you've already done, there are always some caveats. What I do here will help you imagine these things and look around, and you put them into the hands and you see how big they are in operation. It's huge and they'll be hand weighed and they'll be right there. The sense of responsibility we take for ourselves in the frame of mind that we are being taught about, and expect, to be really at odds with that of so many others. Most of the time, you can make good decisions not to have one, the time, the budget, the really. But I believe it sure makes a great choice to do it, I consider it a strategy for going through the high points of your personal circle in your life in that you show things you already done there. "The days that reach is on you."

· When it comes time to finish the week, you read the regular stories for a newspaper, or read comments, The People & Stuff. Or read all the writing, The Fowls, a board reading pinned to something Very Important. If, for instance, as I read the stories in the morning, there was a willingness to put the, just put down and a small willingness to try to just stick with them. I got a loss at blocking or whatever. It was the board reading. "Hey, get down on my knees." "Always don't quit." I don't. If you've pulled yourself under from behind then you don't tend to see things through these modes of thought.

Once you decide that's the challenge here, I myself very, very was invited by this. And so, here's a visit to my teacher in college. It was the night when I went, and it's the one I remember most vividly. The Professor sat me face down, his hand on my shoulder, and said: "Take care of yourself." And said that, there's no bad example. "I know this the only way to get an absolute best understanding of that." To think of my life outside the classroom, when I hadn't gone to college, it was finally done over.

· My wife used to be a college student, and as well as I, I believed in it when I went back to college. Same places where I used to have a chance interview with my spouse, I told myself I'm not poor experience in college anyway, and it ended up helping. I've always enjoyed finding a voice in a tough time, but I still had to go to. I had to settle down. I sorta did the whole thing.

Once I moved away and I found out that a job you can do in your own house isn't even moving around, I just took it. . . It out the window well. I've often been a calm, truly relaxed person. I've tried to be better, and maybe just leave that behind.

In the late months of the school year, it's very important to put your mind in a work mode before your move back to the City. Because, sometimes the road ahead is all before you when you decide how to sleep. Is this in your control.<|endoftext|>

Figure 13: Text sample generated by VADD trained on OpenWebText with 1024 sampling steps.

Table 8: Perplexities on the test split of OpenWebText. All the models are trained for 1M iterations with a batch size of 512 and a context size of 1024. All model sizes correspond to GPT-2 small.

| Model | Perplexity |
|---|---|
| AR (reprduced by Sahoo et al. (2024)) | 17.54 |
| SEDD (reproduced by Sahoo et al. (2024)) | 24.10 |
| MDLM (reported by Sahoo et al. (2024)) | 23.21 |
| MDLM (reproduced by us) | 23.07 |
| VADD ($K = 1$) | 22.56 |
| VADD ($K = 10$) | 22.33 |
| VADD ($K = 50$) | 22.36 |
| VADD ($K = 100$) | 22.27 |
| VADD ($K = 1000$) | **22.21** |

Table 9: Generative perplexity of the model trained on OpenWebText. The results are averaged over 256 independent samples. The categorical sampling is done under fp64 precision.

| Number of sampling steps ($T$) | 16 | 32 | 64 | 128 | 256 | 512 | 1024 |
|---|---|---|---|---|---|---|---|
| MDLM (reproduced by us) | 330.39 | 191.96 | 141.23 | 124.12 | 115.12 | 108.87 | 104.58 |
| VADD | **194.08** | **124.23** | **98.82** | **89.24** | **85.93** | **82.11** | **78.27** |

