# OpenReview forum: "Variational Autoencoding Discrete Diffusion with Enhanced Dimensional Correlations Modeling"
_ICLR.cc/2026/Conference — ICLR 2026 Poster_

### Official Review · Reviewer_Dd6V · 2025-10-30

**Soundness:** 3
**Presentation:** 3
**Contribution:** 2
**Rating:** 4
**Confidence:** 3

**Summary:**

The paper proposes VADD, a generative framework combining discrete diffusion models  with a variational continuous latent variable. The goal is to address the limitations of purely discrete diffusion by introducing a latent variable z that captures global structure, while the discrete diffusion handles token-level generation. The joint training objective is derived as a variational ELBO decomposition into generative model and recognition model.

**Strengths:**

1.	This paper presents a novel idea that introduces latent space modelling to discrete diffusion models, with a VAE-type learning objective formulation.
2.	Experiment results on image and text generation demonstrate the effectiveness of the proposed model.

**Weaknesses:**

1. The motivation for introducing a latent variable into the discrete diffusion model is not sufficiently justified. As discussed in Paragraph 2 of Section 1, the issue with MDMs becomes more pronounced when the number of denoising steps is small. However, if a sufficient number of denoising steps are used, would incorporating a latent variable still offer any advantage in terms of accuracy or efficiency? The paper provides no analysis or evidence to support this claim.

2. For image generation, how is the performance comparison with SOTA continuous diffusion models? The comparison is conducted with only non-latent space modeling, which seems not sufficient.

3. In Section 4, a couple of related works are mentioned. Although there is difference between them and the proposed VADD model,  they need to be incorporated as baselines.

4. No code is released for reproducibility checking.

5. For text generation task, how is the latent space defined?

6. Several symbols like “Cat” are not defined in the paper.

**Questions:**

Please refer to the "Weaknesses" section.

---

> ### Author Response · Authors · 2025-11-26
> **Thanks for your feedback!**
>
> Thanks for your careful review and constructive feedback! We addressed your concerns as follows.
>
> **Q1**. The motivation for introducing a latent variable into the discrete diffusion model is not sufficiently justified. As discussed in Paragraph 2 of Section 1, the issue with MDMs becomes more pronounced when the number of denoising steps is small. However, if a sufficient number of denoising steps are used, would incorporating a latent variable still offer any advantage in terms of accuracy or efficiency? The paper provides no analysis or evidence to support this claim.
>
> **A1**. The dimensional independence issue is more pronounced when the number of denoising steps is small, and therefore, we only expect a significant performance gain with a small number of steps. To see this, note that for masked diffusion models, the number of tokens decoded in a denoising step is proportional to the noise step size $\alpha_t-\alpha_s$. This means that under a smaller number of steps, there are more tokens to be generated simultaneously, making it necessary to model the dimensional correlations.
> In the ideal case of sufficiently many denoising steps, the MDMs will degenerate to any-order autoregressive models, which decode one token at a time; in such a case, modeling the dimensional correlations will become totally meaningless.
> Therefore, we would not expect superior performance of VADD when a sufficient number of denoising steps is used.
>
>
> **Q2**. For image generation, how is the performance comparison with SOTA continuous diffusion models? The comparison is conducted with only non-latent space modeling, which seems not sufficient.
>
> **A2**. Thanks for this suggestion. We'd like to emphasize that the discrete and continuous modeling of images has intrinsically different difficulties, and we generally do not expect the discrete diffusion models to perform comparably to continuous diffusion models on image generation tasks.
> We consider the image generation task here mainly to provide a high-dimensional numerical example, instead of to compare or even beat the continuous diffusion models.
> Despite this, we'd like to provide a DDPM baseline with small NFEs as follows.
> We use the diffusers.DDPMPipeline package and the google/ddpm-cifar10-32 pre-trained checkpoint for sampling from DDPM.
> Similarly to MDLM and VADD, the FID of DDPM is computed using the clean-fid package based on 50K samples.
> We see that when NFE is small, the FID score of DDPM lies between VADD and MDLM.
>
>
> |T | DDPM | MDLM| VADD |
> | ---| ---  |            ---|   ---|
> | 10 | 298.7|334.3          |170.3 |
> | 20 | 239.1|261.3          |108.7 |
> | 30 | 187.2|203.4          |84.8  |
> | 40 | 153.6|166.1          |72.1  |
> | 50 | 131.0|140.3          |64.6  |
> |100 | 77.2 |76.5           |50.5  |
>
>
>
>
>
> **Q3**. In Section 4, a couple of related works are mentioned. Although there is difference between them and the proposed VADD model, they need to be incorporated as baselines.
>
> **A3**. We would like to clarify that we are the first to consider using the latent variable model for modeling the dimensional correlations in discrete diffusion models.
> In fact, the latent variable model can be a quite general idea in generative modeling, and we listed several notable applications as related works.
> Some of these models [Bowman
> et al., 2015; Gu et al., 2018; Kaiser et al., 2018; Kong et al. (2025)] are proposed not for discrete diffusion models but for machine translation, autoregressive generation, etc.
> Although Hayakawa et al. (2024) also investigate improving MDMs, their method relies on a powerful pretrained teacher model, making the comparison to VADD unfair.
>
>
> **Q4**. No code is released for reproducibility checking.
>
> **A4**. We will release our code upon publication.
>
> **Q5**. For text generation task, how is the latent space defined?
>
> **A5**. For the text generation task, the  prior distribution for latent variables $p(z)$ is a standard Gaussian distribution.
>
> **Q6**. Several symbols like “Cat” are not defined in the paper.
>
> **A6**. Sorry for the missing definition. "Cat" refers to the categorical distribution.

---

### Official Review · Reviewer_qhUA · 2025-11-02

**Soundness:** 3
**Presentation:** 3
**Contribution:** 3
**Rating:** 6
**Confidence:** 3

**Summary:**

This paper proposed Variational Autoencoding Discrete Diffusion (VADD), which is a framework that augments masked discrete diffusion models (MDMs) with a latent variable to better capture the correlations between different dimensions. Specifically, the transition probability $p_{\theta}(x_s | x_t)$ in the backward step is parametrized as the marginal of the product distribution between prior distribution $p(z)$ of the latent variable and the conditional denoising distribution $p_{\theta}(x_s | x_t, z)$ given the latent variable $z$. As the marginal is generally intractable, the authors introduced a “double ELBO” (DELBO) framework to jointly train an auxiliary recognition model $r_{\phi}(z | x_s, x_t)$ that approximates the distribution $p_{\theta}(z | x_0, x_t)$ and the conditional denoiser $p_{\theta}(x_0 | x_t, z)$. Experiments on 2D multimodal toy distributions, pixel-level image generation on MNIST and CIFAR, as well as text generation via MDMs are provided to justify the effectiveness of proposed method, which implies that adding an amortized latent to discrete diffusion does improve the quality of samples generated by MDMs.

**Strengths:**

The reviewer finds both the presentation and the motivation of the paper to be crystal clear. To the best of the reviewer's knowledge, this is probably the first work that tries to address the problem of capturing the correlation between different dimensions via the latent variable structure in MDMs, which is motivated from existing work [1] that uses the latent variable structure for next token prediction in autoregressive models. Details about the choice of hyperparameters in the experiments are also included.

**Weaknesses:**

(1) For the image generation task, this paper only tested the proposed methodology on relatively small-scale datasets like MNIST and CIFAR10. In contrast, existing work [2,3] applying MDMs and related variants to image generation have tested on larger-scale datasets like ImageNet 256. The authors are encouraged to include an extra set of experiments on large-scale image datasets like ImageNet to better illustrate the effectiveness of the proposed methodology. (For instance, the authors may refer to the experimental setting of [3].)

(2) Given that the method proposed in this work can be categorized as a training-based type of method for improving the quality of the samples generated by MDMs, the authors might need to perform a more detailed literature review on related methods. For instance, when the authors discussed distillation-type of methods as related work at the beginning of section 4, they only mentioned [7] and missed a few other concurrent work like [4,5,6]. Also, the quality of the paper can be further improved by adding extra training-based baselines (like those based on distillation) to justify the effectiveness of VADD.

**Questions:**

For the backward transition probability parametrized by the latent variable defined in equation (6), would it possible to make the prior distribution of the latent variable $z$ dependent on the time variable $t$? The reviewer thinks that it might be more reasonable to do so, as the correlation between different dimensions also varies as time changes. (Otherwise, would it be possible for the authors to comment on why it makes sense to use the same prior distribution for all time $t$ here?)

Moreover, it seems that the authors only tested the effect of VADD for MDMs. Would it be possible for the authors to comment on/briefly discuss whether the same idea of using latent variable models can also be applied to the case of uniform discrete diffusion models or not?

Overall, the reviewer thinks that the paper might be considered for top ML venues like ICLR, but the authors should probably address all questions above, add papers listed below as extra references and discuss them appropriately.

References:

[1] Kong, Deqian, Minglu Zhao, Dehong Xu, Bo Pang, Shu Wang, Edouardo Honig, Zhangzhang Si et al. "Scalable language models with posterior inference of latent thought vectors." arXiv e-prints (2025): arXiv-2502.

[2] Hayakawa, Satoshi, Yuhta Takida, Masaaki Imaizumi, Hiromi Wakaki, and Yuki Mitsufuji. "Demystifying MaskGIT Sampler and Beyond: Adaptive Order Selection in Masked Diffusion." arXiv preprint arXiv:2510.04525 (2025).

[3] Li, Tianhong, Huiwen Chang, Shlok Mishra, Han Zhang, Dina Katabi, and Dilip Krishnan. "Mage: Masked generative encoder to unify representation learning and image synthesis." In Proceedings of the IEEE/CVF Conference on Computer Vision and Pattern Recognition, pp. 2142-2152. 2023.

[4] Fu, Feiyang, Tongxian Guo, and Zhaoqiang Liu. "Learnable Sampler Distillation for Discrete Diffusion Models." arXiv preprint arXiv:2509.19962 (2025).

[5] Zhu, Yuanzhi, Xi Wang, Stéphane Lathuilière, and Vicky Kalogeiton. "Di [M] o: Distilling masked diffusion models into one-step generator." In Proceedings of the IEEE/CVF International Conference on Computer Vision, pp. 18606-18618. 2025.

[6] Zhu, Yuanzhi, Xi Wang, Stéphane Lathuilière, and Vicky Kalogeiton. "Soft-di [m] o: Improving one-step discrete image generation with soft embeddings." arXiv preprint arXiv:2509.22925 (2025).

[7] Hayakawa, Satoshi, Yuhta Takida, Masaaki Imaizumi, Hiromi Wakaki, and Yuki Mitsufuji. "Distillation of discrete diffusion through dimensional correlations." arXiv preprint arXiv:2410.08709 (2024).

---

> ### Author Response · Authors · 2025-11-26
> **Thanks for your feedback! (part 1/2)**
>
> Thanks for your careful review and constructive feedback! We addressed your concerns as follows.
>
> **Q1**. For the image generation task, this paper only tested the proposed methodology on relatively small-scale datasets like MNIST and CIFAR10. In contrast, existing work [2,3] applying MDMs and related variants to image generation have tested on larger-scale datasets like ImageNet 256. The authors are encouraged to include an extra set of experiments on large-scale image datasets like ImageNet to better illustrate the effectiveness of the proposed methodology. (For instance, the authors may refer to the experimental setting of [3].)
>
>
> **A1**. Thanks for your suggestion.
> We would like to clarify that our model is not designed for image generation, but for a general discrete generation framework with emphasis on the text generation task with a few NFEs.
> We consider the image generation benchmark just for a simple demonstration, with no intention to beat the other image generation benchmarks.
> The large-scale generation performance of VADD is mainly demonstrated by the experiments on the OpenWebText data sets.
> Our experiment design mainly follows MDLM (Sahoo et al., 2024), SEDD (Lou et al., 2024), and RADD (Ou et al., 2025) papers.
>
>
>
> **Q2**. Given that the method proposed in this work can be categorized as a training-based type of method for improving the quality of the samples generated by MDMs, the authors might need to perform a more detailed literature review on related methods. For instance, when the authors discussed distillation-type of methods as related work at the beginning of section 4, they only mentioned [7] and missed a few other concurrent work like [4,5,6]. Also, the quality of the paper can be further improved by adding extra training-based baselines (like those based on distillation) to justify the effectiveness of VADD.
>
> **A2**. Thank you so much for providing these related works.
> Di[M]O [5] distills a multi-step masked diffusion model into a one-step generator, by optimizing the the model outputs of a student model for all possible intermediate states with the help of an auxiliary model.
> Soft-Di[M]O [6] integrates soft embeddings into Di[M]O distillation, which makes one-step generators end-to-end trainable and enables straightforward application of GAN-based refinement, differentiable reward fine-tuning, and TTEO.
> Learnable Sampler Distillation [4] also employs a distillation approach in which a student sampler with a few steps learns to align its intermediate score trajectory with that of a high-quality teacher sampler with numerous steps.
> Compared to these methods, VADD does not need a pre-trained teacher model and thus is more like a novel MDM paradigm instead of a distillation method.
> Moreover, VADD is different from these methods in that it aims to solve the dimensional independence issue.
> Therefore, we think it is unfair to compare our model to these distillation-based methods that rely on powerful pretrained teacher models, which themselves can already be distilled MDMs (e.g., SDTT).
> We have now included these discussions in the related work section in our revision.
> Sorry for missing some of those related papers before, as we note that several of them (e.g., [2][4][6]) are published on arXiv near or after the ICLR submission deadline.
>
>
> **Q3**. For the backward transition probability parametrized by the latent variable defined in equation (6), would it possible to make the prior distribution of the latent variable dependent on the time variable? The reviewer thinks that it might be more reasonable to do so, as the correlation between different dimensions also varies as time changes. (Otherwise, would it be possible for the authors to comment on why it makes sense to use the same prior distribution for all time here?)
>
> **A3**. Thanks for your question! In the current framework, although $p(z)$ does not rely on $t$, the time-dependency can also be modeled by the conditional denoising model $p(x_s|x_t,z)$ which depends on $t$.
> More specifically, say if we want to introduce a time-dependent prior $p_t(z)$ through a time-dependent map $g_t$ of standard gaussian, that is $z_t \sim p_t(z) \Leftrightarrow z_t = g_t(z), \quad z\sim \mathcal{N}(0,I)$. This can be equivalently done by still using the time-independent standard Gaussian prior $p(z) = \mathcal{N}(0,I)$, but adding $g_t$ to the conditional denoising model as $p(x_s|x_t, g_t(z))$, which is known as the reparameterization trick.
> Therefore, we can still expect it work fairly well in the current framework.
> But we acknowledge that designing a more complex prior sample is a meaningful future direction, e.g., $p(z|x_t,t)$ which depends on both $x_t$ and $t$.
> This more informed prior distribution will help capture the correlations among different tokens which can depend on the precedent state $x_t$ and the time variable $t$.
> We have added these discussion in the future work section in our revision.

---

> ### Author Response · Authors · 2025-11-26
> **Thanks for your feedback! (part 2/2)**
>
> **Q4**. Moreover, it seems that the authors only tested the effect of VADD for MDMs. Would it be possible for the authors to comment on/briefly discuss whether the same idea of using latent variable models can also be applied to the case of uniform discrete diffusion models or not?
>
> **A4**. Thanks for this insightful question! Our VADD can also be adapted to uniform discrete diffusion models.
> To see this, note that in the original formulation in D3PM (https://arxiv.org/pdf/2107.03006), the training objective for both uniform and masked discrete diffusion models is $L(\theta) = \mathbb{E}\_{x_t,x_{t-1},x_0} \log p_\theta(x_{t-1}|x_t) $, where $p_\theta(x_{t-1}|x_t)$ is the one-step denoising model. By introducing a conditional denoising model $p_\theta(x_{t-1}|x_t,z)$ and a recognition model $r_\phi(z|x_{t-1},x_t)$, this loss can be transformed into $L(\theta, \phi) = \mathbb{E}\_{x_t,x_{t-1},x_0} \mathbb{E}\_{r_\phi(z|x_{t-1},x_t)}\log (\frac{p_\theta(x_{t-1}|x_t,z)p(z)}{r_\phi(z|x_{t-1},x_t)})$.
>
>
> References:
>
> [1] Kong, Deqian, Minglu Zhao, Dehong Xu, Bo Pang, Shu Wang, Edouardo Honig, Zhangzhang Si et al. "Scalable language models with posterior inference of latent thought vectors." arXiv e-prints (2025): arXiv-2502.
>
> [2] Hayakawa, Satoshi, Yuhta Takida, Masaaki Imaizumi, Hiromi Wakaki, and Yuki Mitsufuji. "Demystifying MaskGIT Sampler and Beyond: Adaptive Order Selection in Masked Diffusion." arXiv preprint arXiv:2510.04525 (2025).
>
> [3] Li, Tianhong, Huiwen Chang, Shlok Mishra, Han Zhang, Dina Katabi, and Dilip Krishnan. "Mage: Masked generative encoder to unify representation learning and image synthesis." In Proceedings of the IEEE/CVF Conference on Computer Vision and Pattern Recognition, pp. 2142-2152. 2023.
>
> [4] Fu, Feiyang, Tongxian Guo, and Zhaoqiang Liu. "Learnable Sampler Distillation for Discrete Diffusion Models." arXiv preprint arXiv:2509.19962 (2025).
>
> [5] Zhu, Yuanzhi, Xi Wang, Stéphane Lathuilière, and Vicky Kalogeiton. "Di [M] o: Distilling masked diffusion models into one-step generator." In Proceedings of the IEEE/CVF International Conference on Computer Vision, pp. 18606-18618. 2025.
>
> [6] Zhu, Yuanzhi, Xi Wang, Stéphane Lathuilière, and Vicky Kalogeiton. "Soft-di [m] o: Improving one-step discrete image generation with soft embeddings." arXiv preprint arXiv:2509.22925 (2025).
>
> [7] Hayakawa, Satoshi, Yuhta Takida, Masaaki Imaizumi, Hiromi Wakaki, and Yuki Mitsufuji. "Distillation of discrete diffusion through dimensional correlations." arXiv preprint arXiv:2410.08709 (2024).

---

> > ### Comment · Reviewer_qhUA · 2025-11-26
> > **Response**
> >
> > The reviewer would like to thank the authors for the detailed response, which has addressed most of my concerns. Therefore, I would like to keep my acceptance score.

---

> > > ### Author Response · Authors · 2025-11-26
> > > **Thanks!**
> > >
> > > Dear Reviewer qhUA,
> > >
> > > Thank you again for your constructive feedback and the positive evaluation of our work!

---

### Official Review · Reviewer_UPfa · 2025-11-03

**Soundness:** 3
**Presentation:** 3
**Contribution:** 3
**Rating:** 8
**Confidence:** 4

**Summary:**

The authors propose the Variational Autoencoding Discrete Diffusion (VADD) framework which augments masked diffusion models (MDM) with latent variable modeling (using an auxiliary recognition model) for implicitly capturing the dependencies between different tokens when unmasking during the denoising process. This results in a more expressive unmasking framework in the low timestep regime. Empirical results show that VADD outperforms the MDM framework on different benchmarks in the low timestep regime.

**Strengths:**

1. I specifically like the problem formulation in the paper. The idea of using latent variables to model the dependencies between different dimensions is quite straightforward yet elegant.

2. The paper is well written in the sense that the method is easy to follow and the empirical results establish the advantages of the method over the baseline MDLM model.

**Weaknesses:**

**Looseness of the Double ELBO (DELBO) bound**: Can the authors comment on how loose is the bound in Eq. 9 compared to the original log likelihood i.e. log p(x_0) and add more intuition in the main text. This could be quite important for applications where metrics like bits per dim (bpd) are quite important.

**Missing Related Work**: The related work section in the paper is extremely limited. Can the authors discuss more about related work in discrete diffusion models, VAEs and posterior collapse. There is a large body of work in discrete diffusion which might be complimentary to the framework introduced by the authors. However an extensive discussion is currently missing from the paper.

**Re. Empirical Results**:

1. Can the authors add continuous diffusion based baselines for image generation for CIFAR-10? Based on my understanding the image quality in Table 3 is rather bad for CIFAR-10 and continuous time methods do much better on this benchmark. In light of this maybe these results can be moved to the Appendix altogether and the remaining space could be utilized for adding more intuitions on the theoretical results in the paper. Moreover, I would request the authors to include qualitative results for CIFAR-10 in the Appendix for a better sense on the quality of the samples.

2. Do the authors have insights about the scaling properties of VADD with model size in terms of perplexities for the text benchmarks?

**Discussion of limitations**: There is limited discussion on the limitations of the methods. In my understanding enforcing the recognition model to lie close to the prior would most likely suffer from prior hole problems since the underlying recognition model can be quite complex. In general, the proposed framework would inherit the same problems that classical VAE based approaches suffer from. Therefore, I think its quite important to discuss these limitations in the main text.

**Questions:**

See weaknesses

---

> ### Author Response · Authors · 2025-11-26
> **Thanks for your feedback! (1/3)**
>
> Thanks for your careful review and constructive feedback! We addressed your concerns as follows.
>
> **Q1**. Looseness of the Double ELBO (DELBO) bound: Can the authors comment on how loose is the bound in Eq. 9 compared to the original log likelihood i.e. $\log p(x_0)$ and add more intuition in the main text. This could be quite important for applications where metrics like bits per dim (bpd) are quite important.
>
> **A1**. Thanks for this question. Firstly, the DELBO $\hat{\mathcal{L}}(x_0;\theta,\phi)$ is a looser lower bound of the log-likelihood compared to the ELBO $\hat{L}(x_0;\theta)$. The gap between them is given by $\int_0^1 \frac{-\alpha_t'}{1-\alpha_t}E_{r_{\phi}(z|x_0,x_t)} KL(r_{\phi}(z|x_0,x_t)|q_{\theta}(z|x_0,x_t)) dt$, which is determined by the fitness of $r_{\phi}(z|x_0,x_t)$.
> DELBO equals to ELBO if and only if $r_{\phi}(z|x_0,x_t) = q_{\theta}(z|x_0,x_t)$.
> Secondly, DELBO becomes tighter by employing the multi-sample lower bound $\hat{\mathcal{L}}_K(x_0;\theta,\phi)$ in equation (20).
> We set $K=1000$ when computing the bpd and perplexity, as stated in the first paragraph in Experiments.
> See an ablation study on different choices of $K$ in Table 7.
>
> **Q2**. Missing Related Work: The related work section in the paper is extremely limited. Can the authors discuss more about related work in discrete diffusion models, VAEs and posterior collapse. There is a large body of work in discrete diffusion which might be complimentary to the framework introduced by the authors. However an extensive discussion is currently missing from the paper.
>
> **A2**. Thanks you for your suggestion! We provide a more detailed discussion in the ``related works'' section.
>
> - Regarding the discrete diffusion models. Di[M]O [5] distills a multi-step masked diffusion model into a one-step generator by optimizing the model outputs of a student model for all possible intermediate states with the help of an auxiliary model.
> Soft-Di[M]O [6] integrates soft embeddings into Di[M]O distillation, which makes one-step generators end-to-end trainable and enables straightforward application of GAN-based refinement, differentiable reward fine-tuning, and TTEO.
> Learnable Sampler Distillation [4] also employs a distillation approach in which a student sampler with a few steps learns to align its intermediate score trajectory with that of a high-quality teacher sampler with numerous steps.
> Our approach is clearly distinct from these distillation-based methods as we train the latent-variable denoising model from scratch based on the training data set, and do not rely on a powerful pre-trained teacher model as the initialization weight
> and learning target.
>
> - Regarding the VAE posterior collapse. The posterior collapse issue is a commonly encountered problem during training VAEs, i.e., the recognition model ignores the dependency on data and becomes very close to the prior distribution.
> From an optimization perspective, the recognition model can be less regularized by the prior through downweighting the KL divergence term, known as the KL annealing strategy [6,7].
> As the posterior collapse issue can be partially attributed to the over-complex decoder, [8] proposes a hybrid architecture that combines a VAE with a weaker decoder, while [9] proposes to strengthen the encoder.
> [10] adds skip connections from the input directly to the decoder, reducing the burden on the latent variable.
> [11] uses normalizing flows to model a highly flexible and complex prior, allowing the posterior to capture complex data dependencies instead of collapsing to the simple prior.
>
> - Regarding the prior hole problem. Please refer to our response to Q5. We have added a detailed discussion in the ``Limitations and future works'' section.

---

> ### Author Response · Authors · 2025-11-26
> **Thanks for your feedback! (2/3)**
>
> **Q3**. Can the authors add continuous diffusion based baselines for image generation for CIFAR-10? (..)
>
>
> **A3**: Thanks for your suggestion! We use the diffusers.DDPMPipeline package and the google/ddpm-cifar10-32 pre-trained checkpoint for sampling from DDPM.
> Similarly to MDLM and VADD, the FID of DDPM is computed using the clean-fid package based on 50K samples.
> We see that when NFE is small, the FID score of DDPM lies between VADD and MDLM.
> The reason VADD performs the best in this small NFE regime is that VADD enables direct correlation modeling via the variational autoencoding latent variable structure, while other baseline methods only rely on the composition of repetitive function evaluations to capture correlations and would, in general, require a large NFE to produce high-quality samples.
> We also provide the CIFAR-10 image samples from VADD with different sampling steps in Figure 9 in the appendix.
>
> |T | DDPM | MDLM| VADD |
> | ---| ---  |            ---|   ---|
> | 10 | 298.7|334.3          |170.3 |
> | 20 | 239.1|261.3          |108.7 |
> | 30 | 187.2|203.4          |84.8  |
> | 40 | 153.6|166.1          |72.1  |
> | 50 | 131.0|140.3          |64.6  |
> |100 | 77.2 |76.5           |50.5  |
>
> **Q4**. Do the authors have insights about the scaling properties of VADD with model size in terms of perplexities for the text benchmarks?
>
> **A4**. Thanks for your question! In our opinion, VADD introduces an additional scaling dimension - the dimension of the hidden variable $z$.
> By introducing the latent variable $z$, we expect a high contribution margin when increasing the hidden dimension.
> Note that increasing the dimension of the latent variable would not change the model size, regardless of the minor change in the input dimension.
> Regarding the scaling low on the model size, we expect VADD has a similar result with other MDMs, e.g., the scaling curves w.r.t. model size and data size (https://arxiv.org/pdf/2308.12219).
>
>
> **Q5**. Discussion of limitations: There is limited discussion on the limitations of the methods. (..)
>
> **A5**. Thank you so much for pointing this out, and we have added the following discussion in the newly added ``limitations and future works'' section.
> A prior hole refers to regions that have high probability under the VAE's prior but low probability under the VAE's posterior.
> As we used an uninformed Gaussian prior for the latent variable $p(z)$, VADD can also suffer from this problem and produce poor-quality samples.
> A meaningful future direction is to consider an alternative denoising distribution $p_\theta(x_s|x_t)=\int p_\theta(x_s|x_t,z)p_\theta(z|x_{t})d z$, where we use a more informed prior distribution $p_\theta(z|x_{t})$ that depends on the partially masked sample $x_t$.
> More complex structures, e.g., hierarchical priors, can also be assumed for $p_\theta(z|x_{t})$.
> See [1][2] for VAE learning procedures that alleviate the prior hole problem.

---

> > ### Author Response · Authors · 2025-11-26
> > **Thanks for your feedback! (3/3)**
> >
> > ## References
> > [1] Aneja J, Schwing A, Kautz J, et al. A contrastive learning approach for training variational autoencoder priors[J]. Advances in neural information processing systems, 2021, 34: 480-493.
> >
> > [2] Hoffman M D, Johnson M J. Elbo surgery: yet another way to carve up the variational evidence lower bound[C]//Workshop in Advances in Approximate Bayesian Inference, NIPS. 2016, 1(2).
> >
> > [3] Fu, Feiyang, Tongxian Guo, and Zhaoqiang Liu. "Learnable Sampler Distillation for Discrete Diffusion Models." arXiv preprint arXiv:2509.19962 (2025).
> >
> > [4] Zhu, Yuanzhi, Xi Wang, Stéphane Lathuilière, and Vicky Kalogeiton. "Di [M] o: Distilling masked diffusion models into one-step generator." In Proceedings of the IEEE/CVF International Conference on Computer Vision, pp. 18606-18618. 2025.
> >
> > [5] Zhu, Yuanzhi, Xi Wang, Stéphane Lathuilière, and Vicky Kalogeiton. "Soft-di [m] o: Improving one-step discrete image generation with soft embeddings." arXiv preprint arXiv:2509.22925 (2025).
> >
> > [6] Samuel R Bowman, Luke Vilnis, Oriol Vinyals, Andrew M Dai, Rafal Jozefowicz, and Samy Bengio. Generating sentences from a continuous space. arXiv preprint arXiv:1511.06349, 2015
> >
> > [7] Zichao Yang, Zhiting Hu, Ruslan Salakhutdinov, and Taylor Berg-Kirkpatrick. Improved variational autoencoders for text modeling using dilated convolutions. In International conference on machine learning, pp. 3881–3890. PMLR, 2017
> >
> > [8] Ishaan Gulrajani, Kundan Kumar, Faruk Ahmed, Adrien Ali Taiga, Francesco Visin, David Vazquez, and Aaron Courville. Pixelvae: A latent variable model for natural images. In International Conference on Learning Representations, 2017.
> >
> > [9] Yoon Kim, Sam Wiseman, Andrew Miller, David Sontag, and Alexander Rush. Semi-amortized variational autoencoders. In International Conference on Machine Learning, pp. 2678–2687. PMLR, 2018
> >
> > [10] Adji B Dieng, Yoon Kim, Alexander M Rush, and David M Blei. Avoiding latent variable collapse with generative skip models. In The 22nd International Conference on Artificial Intelligence and Statistics, pp. 2397–2405. PMLR, 2019.
> >
> > [11] Durk P Kingma, Tim Salimans, Rafal Jozefowicz, Xi Chen, Ilya Sutskever, and Max Welling. Improved variational inference with inverse autoregressive flow. Advances in neural information processing systems, 29, 2016.

---

### Official Review · Reviewer_ZCGk · 2025-11-04

**Soundness:** 2
**Presentation:** 2
**Contribution:** 2
**Rating:** 4
**Confidence:** 4

**Summary:**

This paper proposes variational autoencoding discrete diffusion to enhance the dimensional correlation modeling. Along with introducing additional recognition model, they also propose some adaptions to architecture design. The experiment shows that the model achieve better FID than MDLM in text and image modeling.

**Strengths:**

1. The paper is well-organized and easy to follow.

2. The performance of model outperforms other discrete diffusion model in pixel and language tasks.

**Weaknesses:**

1. The paper does not technically sound to me. It is not clear why adding latent variable $z$ contribute to model performance since in the sampling, $z$ is simply sampled from prior $p(z)$ distribution without using recognition model. $z$ seems like a style vector like StyleGAN or DDGAN which just help to increase the stochasticity of the architecture.

2. Why not using the train recognition model for other iteration $i < T$ like hierarchical latent VAE technique. It seems a wasteful for training   recognition model.

3. The performance on pixel space seems marginal.

4. The proposed architecture is simple for denoising model, and does not show much novelty and lacks ablation study on different way to integrate $z$ into the model architecture in both denoising and recognition model.

**Questions:**

See that weakness

---

> ### Author Response · Authors · 2025-11-26
> **Thanks for your feedback!**
>
> Thanks for your careful review and constructive feedback! We addressed your concerns as follows.
>
> **Q1**. The paper does not technically sound to me. It is not clear why adding latent variable contribute to model performance since in the sampling, is simply sampled from prior distribution without using recognition model. $z$ seems like a style vector like StyleGAN or DDGAN which just help to increase the stochasticity of the architecture.
>
> **A1**. This sampling procedure strictly follows the probability definition of the denosing model $p(x_s|x_t)=\int p(x_s|x_t,z)p(z)d z$, where $z$ contributes to the model performance by modeling the correlations among tokens and improving model capacity, even if $z$ is sampled from a prior model.
> To see this, (i) the denoising distribution $p(x_s|x_t)$ in VADD is capable of modeling the dimensional correlations by introducing the latent structures, while the vanilla MDMs cannot.
> Intuitively, the latent variable $z$ governs all the dimensions to denoise towards a mode in data distribution in a collaborative way.
> Although $z$ is sampled from an uninformed prior, it is the conditional denoising model $p(x_s|x_t,z)$ that captures the semantic information in $z$.
> (ii) The latent variable structure implicitly defines the denoising model $p(x_s|x_t)$ by the integral (or weighted sum) of multiple conditional denoising models $p(x_s|x_t,z)$, which is a powerful and general way to improve model capacities as in variational autoencoders (VAEs).
>
> **Q2**. Why not using the train recognition model for other iteration $i<T$ like hierarchical latent VAE technique. It seems a wasteful for training recognition model.
>
> **A2**. Sampling from VADD strictly obeys the probability definition of the denoising model $p(x_s|x_t)=\int p(x_s|x_t,z)p(z)d z$, which requires $z$ to be sampled from the prior model.
> Moreover, the recognition model is defined as $r_\phi(z|x_0,x_t)$, where the clean $x_0$ is unknown during the sampling process, making it impossible to sample from the recognition model.
> In the hierarchical VAE model (Figure 2b in https://arxiv.org/pdf/2007.03898), the hierarchical prior $z=(z_1,\ldots, z_L)$ is still sampled from the prior $p(z)=\prod_l p(z_l|z_{<l})$ instead of the recoginition model $q(z|x)$.
> In our opinion, a meaningful improvement over the current methodology is to assume the denoising model as $p(x_s|x_t)=\int p(x_s|x_t,z)p(z|x_t)d z$, where the trainable prior $p(z|x_t)$ brings the information of the partially masked sample $x_t$.
> We've added this discussion to our limitations.
>
> **Q3**. The performance on pixel space seems marginal.
>
> **A3**. We'd like to make the following comments on the minor improvement on bits per dimension (bpd).
> Firstly, the optimal value of bpd is a positive value $b>0$, and the true improvement of VADD should be $bpd_{vadd}- b$ over $bpd_{mdlm}- b$.
> Secondly, as we have discussed in the first paragraph of Experiments, we emphasize that VADD has more advantages in sample-quality-based metrics (e.g., FID) instead of likelihood-based metrics (e.g., bpd). This is because the likelihood based metrics assume sufficiently many NFEs and hence cannot properly reflect the sample quality under the scenario of few NFEs.
>
> **Q4**.The proposed architecture is simple for denoising model, and does not show much novelty and lacks ablation study on different way to integrate into the model architecture in both denoising and recognition model.
>
> **A4**. We'd like to clarify that our major contribution is a general framework of enhancing the dimensional correlations modeling in MDMs with latent variable structure.
> This framework is generally applicable not only to text modeling with transformer architecture, but also to modeling diverse modalities for discrete data.
> Therefore, the specific design of model architecture is not our main concern in this paper.
> We acknowledge that the architecture design for the denoising model for texts is simple and not very novel, but this simple choice works fairly well in our experiments.
> Exploring powerful architecture design for the denoising model is a meaningful future direction, and we will discuss it in our revision.

---

### Author Response · Authors · 2025-11-26
**Global Response (1/2)**

We thank all reviewers for their careful review and constructive feedback. We would like to first make some comments on several points:

- **The role of latent variables**. Compared to vanilla MDMs, we introduce a latent variable structure as well as its optimization as an improvement. The latent variable $z$ contributes to the model performance by modeling the correlations among tokens and improving model capacity, even if $z$ is sampled from a prior model.
To see this, (i) the denoising distribution $p(x_s|x_t)$ in VADD is capable of modeling the dimensional correlations by introducing the latent structures, while the vanilla MDMs cannot.
Intuitively, the latent variable $z$ governs all the dimensions to denoise towards a mode in data distribution in a collaborative way.
Although $z$ is sampled from an uninformed prior, it is the conditional denoising model $p(x_s|x_t,z)$ that captures the semantic information in $z$.
(ii) The latent variable structure implicitly defines the denoising model $p(x_s|x_t)$ by the integral (or weighted sum) of multiple conditional denoising models $p(x_s|x_t,z)$, which is a powerful and general way to improve model capacities as in variational autoencoders (VAEs).

- **The design space of the recognition model**. In the current framework, the prior distribution $p(z)$ is a standard Gaussian distribution $\mathcal{N}(0,I)$.
Although we find this design works fairly well in practice, this prior distribution can be extended by incorporating the dependency on the partially masked sample $x_t$ and the time $t$.
Specifically, by assuming a prior $p(z|x_t,t)$, the denoising distribution can be written as $p(x_s|x_t) = \int p(x_s|x_t,z)p(z|x_t,t) dt$.
This more informed prior distribution will help capture the correlations among different tokens, which can depend on the precedent state $x_t$ and the time variable $t$.


- **Clarification on the marginal improvement on bits per dimension**. To understand this phenomenon, first note that the optimal value of bpd is a positive value $b>0$, and the true improvement of VADD should be $bpd_{vadd}- b$ over $bpd_{mdlm}- b$.
Secondly, as we have discussed in the first paragraph of Experiments, we emphasize that VADD has more advantages in sample-quality-based metrics (e.g., FID) instead of likelihood-based metrics (e.g., bpd). This is because the likelihood based metrics assume sufficiently many NFEs and hence cannot properly reflect the sample quality under the scenario of few NFEs.


- **Comparision with DDPM for image generation**.
We use the diffusers.DDPMPipeline package and the google/ddpm-cifar10-32 pre-trained checkpoint for sampling from DDPM.
Similarly to MDLM and VADD, the FID of DDPM is computed using the clean-fid package based on 50K samples.
We see that when NFE is small, the FID score of DDPM lies between VADD and MDLM.
The reason VADD performs the best in this small NFE regime is that VADD enables direct correlation modeling via the variational autoencoding latent variable structure, while other baseline methods only rely on the composition of repetitive function evaluations to capture correlations and would, in general, require a large NFE to produce high-quality samples.

|T | DDPM | MDLM| VADD |
| ---| ---  |            ---|   ---|
| 10 | 298.7|334.3          |170.3 |
| 20 | 239.1|261.3          |108.7 |
| 30 | 187.2|203.4          |84.8  |
| 40 | 153.6|166.1          |72.1  |
| 50 | 131.0|140.3          |64.6  |
|100 | 77.2 |76.5           |50.5  |

---

> ### Author Response · Authors · 2025-11-26
> **Global Response (2/2)**
>
> We have uploaded a revised manuscript with the following major changes.
>
> - We added the more discussion on related works, including the distillation based acceleration of MDMs and the posterior collapse issue of VAE.
> - We added a Section ``Limitations and future works'' and discussed that the prior distribution is limited to simple Gaussian distribution in the current framework. We also pointed out potential improvement upon this uninformed prior distribution.
> - For the image generation task on CIFAR10, we added the FID results of DDPM under small number of sampling steps, as a baseline of continuous diffusion models. We see that the sample quality of DDPM lies between that of MDLM and VADD.  We also provided CIFAR-10 samples generated by VADD in Figure 9.

---

### Meta-Review · Area_Chair_Dkmv · 2026-01-11

**Summary:**

This paper proposes Variational Autoencoding Discrete Diffusion (VADD), which augments masked discrete diffusion models with a continuous latent variable and recognition model to better capture inter-token correlations, especially in the few-NFE regime. Reviewers generally agreed that the idea is intuitive and potentially useful, providing a global coordination signal for parallel denoising.

The reviews are mixed. Two reviewers were clearly supportive while others were more skeptical, mainly questioning the novelty relative to prior latent-variable or distillation-based methods, the breadth of empirical evaluation, and the strength of evidence on image benchmarks.

Overall, this is a borderline paper: the idea is interesting and well-motivated, but the empirical evidence is not compelling across all settings. I lean weakly toward acceptance, while acknowledging that either decision would be reasonable.

**Reviewer Concerns:**

Addressed by the rebuttal:
1. clarified why the latent variable helps even when sampled from the prior, and why gains are expected mainly in the few-NFE regime
2. added DDPM baselines to better contextualize image generation results

Still outstanding (non-blocking):
1. Image evidence: CIFAR results remain weak relative to continuous diffusion; several reviewers felt they are not strong enough for main text emphasis.
2.  Baseline breadth and ablations: comparisons to recent discrete-diffusion acceleration methods and deeper ablations on latent usage are still limited
3. Recognition model justification: some reviewers remain unconvinced that the added training-time complexity is fully justified by the observed gains
4. Positioning: the paper would benefit from clearer framing around few-step sample quality rather than likelihood improvements

**Reviewer Scores:**

there would be not much change

---

### Decision · Program_Chairs · 2026-01-26

Accept (Poster)